# Estimating total attenuation using Rayleigh targets at cloud top: applications in multi-layer and mixed-phase clouds observed by ground-based multi-frequency radars

Frédéric Tridon[1], Alessandro Battaglia[2,3], and Stefan Kneifel[1]

[1]Institute for Geophysics and Meteorology, University of Cologne, Cologne, Germany
[2]Department of Environment, Land and Infrastructure Engineering, Politecnico di Torino, Torino, Italy
[3]Department of Physics and Astronomy, University of Leicester, Leicester, UK

**Correspondence:** Frédéric Tridon (ftridon@uni-koeln.de)

**Abstract.** At millimeter wavelengths, attenuation by hydrometeors, such as liquid droplets or large snowflakes, is generally not negligible. When using multi-frequency ground-based radar measurements, it is common practice to use the Rayleigh targets at cloud top as a reference in order to derive attenuation-corrected reflectivities and meaningful dual-frequency ratios (DFR). By capitalizing on this idea, this study describes a new quality-controlled approach aiming at identifying regions of the cloud where particle growth is negligible. The core of the method is the identification of a "Rayleigh plateau", i.e. a large enough region near cloud top where the vertical gradient of DFR remains small.

By analyzing co-located $K_a$-W band radar and microwave radiometer (MWR) observations taken at two European sites under various meteorological conditions, it is shown how the resulting estimates of differential path-integrated attenuation ($\Delta$PIA) can be used to characterize hydrometeor properties. When the $\Delta$PIA is predominantly produced by cloud liquid droplets, this technique alone can provide accurate estimates of the liquid water path. When combined with MWR observations, this methodology paves the way towards profiling the cloud liquid water and/or quality flagging the MWR retrieval for rain/drizzle contamination and/or estimating the snow differential attenuation.

*Copyright statement.* TEXT

## 1 Introduction

Clouds and precipitation play a crucial role not only in weather prediction, but also for climate projections, as they have manifold impacts on the radiation and energy budget (IPCC, 2013; Wild et al., 2013; Zelinka et al., 2017), water cycle (Stephens et al., 2012; L'Ecuyer et al., 2015), and large-scale circulation (Houze, 2014). Accurate retrievals of vertical profiles of cloud and precipitation properties from space or from the ground are essential pillars for evaluating and further developing their representation in numerical models (Iguchi and Matsui, 2018). However, a recent study by Duncan and Eriksson (2018) shows for example that even essential columnar cloud properties, such as ice water path (IWP), show large biases between different

retrieval products, which hampers their applicability to further improve model parametrisations. As mentioned by Duncan and Eriksson (2018) and others, one of the main reasons for the spread between retrieval products but also for differences in models, is related to uncertainties in the underlying cloud microphysics.

Cloud and precipitation radars are key components of any observing system aimed at a detailed characterization of the vertical structure of clouds and precipitation. This has been thoroughly demonstrated by the current and past constellation of space-borne atmospheric radars (see the review by Battaglia et al. (2020a)) and by the increased amount of ground-based (e.g., Löhnert et al. (2015); Kollias et al. (2020); Lubin et al. (2020)) and air-borne (e.g. Kulie et al. (2014); Battaglia et al. (2016); Mason et al. (2017); Chase et al. (2018)) facilities employing suites of polarimetric, multi-frequency, and/or Doppler radars.

In order to provide insights into microphysical processes, a single frequency radar is often insufficient. In addition to exploiting synergies with other remote sensors, a combination of different radar frequencies has been shown in the past to substantially improve the quality of retrievals in ice (Leinonen et al., 2018; Mason et al., 2018) or rain (Tridon et al., 2017a; Battaglia et al., 2020b). These retrievals typically utilize the frequency dependence of attenuation or backscattering of various hydrometeors.

Under certain conditions, the differential reflectivity signal can be attributed completely to either differential scattering or attenuation. The differential scattering signal is generally closely related to the characteristic size of a particle size distribution (PSD). Several studies utilized differential scattering signals partly in combination with Doppler information for retrievals of snowfall (Hogan and Illingworth, 1999; Liao and Meneghini, 2011; Matrosov, 2011; Kneifel et al., 2016; Grecu et al., 2018; Leinonen et al., 2018; Mason et al., 2018, 2019; Barrett et al., 2019; Tridon et al., 2019a) and rain (Firda et al., 1999; Tridon and Battaglia, 2015; Williams et al., 2016; Tridon et al., 2017a, b; Matrosov, 2017; Mróz et al., 2020; Tridon et al., 2019b). In other situations, non-Rayleigh scattering effects are negligible and the attenuation signal can be used to retrieve for example cloud liquid water in pure liquid clouds using $K_a$-W band (Hogan et al., 2005; Huang et al., 2009; Zhu et al., 2019) or rainfall using $K_a$-band (Matrosov, 2005; Matrosov et al., 2006). Recent development (Roy et al., 2018; Battaglia et al., 2020a) of radars operating at even higher frequencies, such as G-band (120 to 300 GHz, Battaglia et al. (2014)), will allow to extract even larger attenuation signals in the near future.

In the majority of cases, differential attenuation contributions are not negligible because at least one of the frequency is affected by attenuation (L'Ecuyer and Stephens, 2002). As a result, in the implementation of microphysics profiling algorithms, attenuation profiles must be derived first, so that non-Rayleigh and attenuation contributions can be disentangled. Hitschfeld and Bordan (1954) described a methodology for estimating attenuation directly from measured reflectivities via an iterative process. Such methodology becomes quickly unstable with increasing attenuation (Marzoug and Amayenc, 1994; Iguchi and Meneghini, 1994; L'Ecuyer and Stephens, 2002). In case of vertically pointing Doppler radars, the Doppler spectrum can be used to separate differential attenuation from differential scattering (Tridon et al., 2013a). The rationale of this method is that, even if large particles are present, the small and slow falling particles which scatter in the Rayleigh regime populate a specific part of the spectrum. Hence, their (spectral) reflectivity should be frequency independent and any difference can be attributed to attenuation. Tridon et al. (2017a) used this principle to retrieve PSD and turbulence during rainfall and Li and Moisseev (2019) derived the attenuation characteristics due to the melting layer. However, this technique requires very high quality of the Doppler spectra including a very high accuracy of the radar beam alignment as well as low turbulence conditions. In more

general applications, additional integral constraints such as the path integrated attenuation (PIA) can be used to stabilize the attenuation correction (Haynes et al., 2009; Liao and Meneghini, 2019).

The underpinning idea for any PIA technique is to use "natural targets" whose intrinsic (differential) backscattering characteristics are well defined. Examples include:

1. the surface reference technique (SRT), which exploits the well-behaved backscattering properties of ocean and, to a lesser degree, land surfaces, and is generally applicable to measurements from air-borne and space-borne platforms (Meneghini et al., 2000; Haynes et al., 2009; Meneghini et al., 2015). When several radar frequencies are available, differential SRT-PIA ($\Delta$PIA) estimates have been proved to be even more robust than single frequency estimates (Battaglia et al., 2016; Liao and Meneghini, 2019);

2. the mountain reference technique applicable to ground-based scanning precipitation radars for rays that intersect mountain clutter (Delrieu et al., 1997; Serrar et al., 2000);

3. the Doppler spectral ratio techniques which require radar observations at multiple frequencies and are based on recovering differential attenuation profiles from the spectral power ratios of the Doppler velocities corresponding to the Rayleigh slow-falling particles (Tridon and Battaglia, 2015; Tridon et al., 2017a; Li and Moisseev, 2019).

This study focuses on the third method, i.e. the exploitation of the small targets that backscatter according to Rayleigh law (Bohren and Huffman, 1983) at all radar observing frequencies as tracers of the differential path integrated attenuation $\Delta$PIA. However, this method exploits standard radar moments, does not require to record high-quality Doppler spectra and can, in principle, be also applied to scanning ground-based multi-frequency radars. Previously, the cloud region with potential Rayleigh particles has been identified using thresholds of reflectivity (Hogan et al., 2000; Kneifel et al., 2015; Dias Neto et al.,
2019). Although there is undoubtedly a general correlation between the strength of differential scattering and radar reflectivity (Matrosov et al., 2019), this threshold method also has a number of disadvantages. First, the threshold depends on frequency (lower frequency radars can accept a larger reflectivity threshold) and if a too strict threshold is chosen, the region with potential Rayleigh targets might become very small. In other situations, the concentration of larger particles might be small enough to cause a reflectivity smaller than the threshold, but their differential scattering signal might be non-negligible. The threshold
method also does not apply any quality control on the differential reflectivities themselves, which are often found to be rather noisy (Battaglia et al., 2020b) due to non-perfect backscattering volume matching and possible mispointing of the antenna beams.

In this paper, a rigorous new procedure for deriving $\Delta$PIA from ground-based multi-frequency zenith-pointing radars is presented (description in Sect. 3) and exemplified in the case of the $K_a$-W band pair of radar frequencies. It is then applied to
a multi-layered cloud with an ice cloud on top of a low level liquid cloud (Sect. 4) and a mixed-phase cloud with supercooled liquid layers embedded in an ice cloud (Sect. 5). The impact and the potential benefits/applications of this technique are discussed in Sect. 6.

## 2 Hydrometeor attenuation at millimeter wavelengths

During the past decade, millimeter wavelength (cloud) radars have become essential tools for the observations of clouds and precipitation. Cloud radars provide particular advantages for cloud and precipitation studies due to their narrow beam width, inherent high sensitivity, portability, reduced susceptibility to Bragg scattering and ground clutter (Kollias et al., 2007). These advantages, however, come with the cost of larger signal attenuation caused by atmospheric gases and hydrometeors, which in general increases with frequency. While attenuation mainly limits the maximum range of possible radar observations, the frequency dependent attenuation signal can also be used as source of information. For example, Hogan et al. (2005); Huang et al. (2009); Zhu et al. (2019) used the differential attenuation between $K_a$ and W band to infer cloud liquid water in pure liquid clouds. Similarly, the attenuation signal at $K_a$ band was used by Matrosov (2005); Matrosov et al. (2006) to derive rainfall rate.

For droplets and ice crystals whose sizes remain much smaller than wavelength of millimeter microwave radiation, the Rayleigh approximation (Bohren and Huffman, 1983) is applicable for computing scattering and absorption properties. In this regime, absorption and scattering efficiencies are both related to the size parameter, $x \equiv \frac{\pi D}{\lambda}$ ($D$ is a characteristic size of the target, $\lambda$ the wavelength of the radiation transmitted by the radar), i.e. proportional to $x$ or $x^4$, respectively. But, while absorption is proportional to the imaginary part of the Clausius-Mossotti factor $K = \frac{n^2-1}{n^2+2}$ (where $n$ is the refractive index of the scatterer), scattering is proportional to the absolute value of the square of $K$. The much larger imaginary part of the liquid water versus the ice refractive index explains the generally larger absorption of microwave radiation by liquid hydrometeors as compared to ice particles. Attenuation coefficients, defined as the integral of the absorption cross sections (efficiencies times $\frac{\pi}{4}D^2$) over the PSD, are then proportional to the equivalent water mass per unit volume.

On the other hand, large raindrops/ice crystals, graupel, and hailstones must be generally considered as non-Rayleigh targets at millimeter wavelengths. In first approximation, by treating particles as spheres, electromagnetic scattering computations based on Mie theory can be used (Lhermitte, 1990). More complex computations are generally needed for accurately describing the scattering properties of large raindrops and snowflakes that exhibit non-spherical shapes (an exhaustive review is provided in Kneifel et al. (2020)).

### 2.1 Liquid hydrometeors

The attenuation coefficients per unit mass (hereafter indicated with $k_{em}$ and expressed in dB m$^2$ kg$^{-1}$) for raindrops is shown in Fig. 2 of Battaglia et al. (2014). The starting value at small sizes corresponds to the Rayleigh absorption value for droplets:

$$k_{em}^{cw} = 81.863 \times \frac{Im(-K)}{\lambda} \tag{1}$$

where cw stands for cloud water. The resulting one-way attenuation produced by a 1 km thick liquid cloud with a liquid water content of 1 g m$^{-3}$ (corresponding to a liquid water path of 1 kg m$^{-2}$) is negligible at $K_u$-band and becomes increasingly significant for higher frequencies, up to $\approx 4$ dB km$^{-1}$ at W-band (Fig. 1). The dielectric properties (i.e., refractive indices) of liquid water also depend on temperature. Because laboratory measurements of the refractive index of supercooled liquid

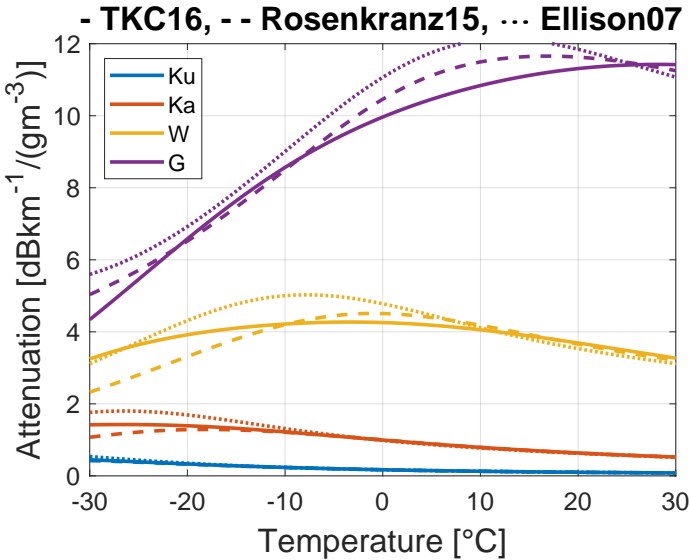

**Figure 1.** Attenuation coefficient of 10 $\mu m$ radius droplets (Rayleigh regime) as function of temperature for the $K_u$ (13.6 GHz), $K_a$ (35 GHz), W(94 GHz), and G-bands (220 GHz). The different line types show the discrepancies between three recent models for computing the liquid water refractive index (Turner et al., 2016; Rosenkranz, 2015; Ellison, 2007).

water are challenging, refractive index models differ more at negative temperatures. This is illustrated for three recent models (Ellison, 2007; Rosenkranz, 2015; Turner et al., 2016) shown in Fig. 1.

For rain (larger drop sizes), the attenuation coefficient steadily increases till a maximum close to $D/\lambda \approx 3$, and then monotonically decreases to a frequency-independent value corresponding to extinction efficiencies of 2 as expected in the geometrical optics limit. Note that the contribution of scattering to attenuation increases with droplet size.

## 2.2   Solid hydrometeors

For ice particles, attenuation is dominated by scattering and is a steadily increasing function of size and density (see Fig. A6 in Battaglia et al. (2020a)). While attenuation by ice crystals is negligible at all frequencies below 200 GHz, snowflakes tend to produce non negligible attenuation at and above W-band. Recent findings by Protat et al. (2019) show that W-band one-way attenuation of the order of 0.5-0.8 dB km$^{-1}$ for reflectivities between 13 and 18 dBZ and up to 2 dB km$^{-1}$ for reflectivities exceeding 20 dBZ can be expected in the ice anvils of tropical convective clouds. Snow attenuation considerably increases when moving from the W to the G-band (Battaglia et al., 2014; Wallace, 1988), reaching one-way values of 0.9, 2.5 and 8.7 dB m$^2$ kg$^{-1}$ at 96, 140, 225 GHz (Nemarich et al., 1988). Graupel (hail) particles found in deep convection produce tangible attenuation already at $K_a$-band ($K_u$-band) as demonstrated in Battaglia et al. (2016).

## 2.3 Melting hydrometeors

Melting particles are generally very efficient in attenuating microwave radiation because they tend to appear as large water particles so that the melting layer not only corresponds to a region of enhanced backscattering (bright band) but also of enhanced extinction (Battaglia et al., 2003), which can account for several dBs of attenuation. Recent observational findings by Li and Moisseev (2019) refined parametrisations of melting layer attenuation at $K_a$ and W-band based on theoretical computations (Matrosov, 2008). As a reference, an equivalent 1 mm h$^{-1}$ (3 mm h$^{-1}$) rainfall is expected to produce an average one-way

specific attenuation of 1.2 (1.9) at $K_a$-band and 3.4 (4.7) dB km$^{-1}$ at W-band during melting. Once melted completely into rain, the attenuation reduces to 0.2 (0.68) at $K_a$-band and 1.43 (3.0) dB km$^{-1}$ at W-band.

## 3 The DFR Rayleigh plateau method

By definition, the PIA monotonically increases with range, or remains constant if the hydrometeor attenuation is negligible. Attenuation leads to a reduction of the measured radar reflectivity and cannot be easily estimated when using a single frequency

radar. Since the attenuation coefficient generally increases with frequency, coincident measurements with an additional radar at a non-attenuating (or less attenuating) frequency provides a reference for determining the $\Delta$PIA between the two frequencies.

When comparing the reflectivities measured (indicated with $Z_m$) by two radars operating at different frequencies $f_1$ and $f_2$ ($f_1 < f_2$), their difference in logarithmic units (expressed in dB and called the dual frequency ratios, DFR($f_1, f_2, r$)) receives contribution from differential (non-Rayleigh) scattering and differential attenuation (Tridon et al., 2013a; Battaglia et al.,

2020a):

$$
\begin{aligned}
\mathrm{DFR}(f_1, f_2, r) & \equiv Z_{m,f_1}(r) - Z_{m,f_2}(r) \\
& = \underbrace{Z_{e,f_1}(r) - Z_{e,f_2}(r)}_{\text{non-Rayleigh}} + \underbrace{2 \int_0^r [k_{em,f_2}(s) - k_{em,f_1}(s)]\, \mathrm{WC}(s)\, ds}_{\text{differential attenuation}}.
\end{aligned}
\tag{2}
$$

where $Z_e$ is the effective reflectivity and WC is the water content in g m$^{-3}$. In this work, the contribution of interest is the differential attenuation (second term of Equation 2): it can be estimated from DFR (where the frequency and range indices have been removed for simplicity) in cloud parts where the non-Rayleigh scattering (first term) is negligible, i.e. where only

small hydrometeors are present. The following analysis will focus on the DFR measured near cloud tops, where the reflectivity is predominantly due to ice particles (with the exception of mixed-phase clouds, as discussed later). Therefore, everywhere hereafter, reflectivities are defined with the convention introduced by Hogan et al. (2006) so that small ice particles (and not small water droplets) have the same $Z_e$.

A simple traditional approach to ensure the presence of only small hydrometeors is to restrict the data to regions where the reflectivity is lower than a certain threshold (for example $Z_e < -10$ dBZ in Dias Neto et al. (2019)). This is based on the assumption that increasing reflectivity is connected to growth processes and hence the presence of increasingly large particles. While this is generally true, it is quite obvious that also a high concentration of small and hence perfect Rayleigh scatterers

could produce a $Z_e$ which is larger than this threshold. An opposite scenario would be a very low number of snow aggregates whose $Z_e$ would be below the threshold but the snowflakes would be far from being Rayleigh targets. From those examples, the problem with a fixed $Z_e$ threshold becomes obvious. The threshold does not only depend on the radar frequency, but also on the details of the particle size distribution, and hence, in principle, has to be adjusted on a case by case basis.

In this work, a more general approach is proposed for estimating $\Delta$PIA from ground-based multi-frequency radar measurements. The DFR is closely related to the characteristic size of the PSD, which is generally expected to increase due to particle growth processes. A DFR threshold would therefore seem to be a more reliable measure for the presence of large non-Rayleigh targets. However, larger DFRs could be also caused by attenuation due to gases and hydrometeors from layers below. As discussed before, Doppler spectra principally allow to disentangle attenuation and scattering effects. However, the spectra at cloud top are in general very narrow, which makes the separation more challenging. Also antenna mispointing effects (different shift of spectra) can be expected to be maximum at high altitudes due to the generally stronger horizontal winds.

In the new approach, ice particles close to cloud top are assumed to be small enough to produce negligible differential scattering and attenuation. As a result, any measured DFR should be a result of path integrated attenuation from the cloud below. When moving downward from cloud top, the DFR remains constant down to the altitude where some ice particles reach sizes which cause non-Rayleigh scattering at the highest frequency used. The goal of this method is therefore to find a plateau of DFR close to cloud top and will be denominated the "DFR Rayleigh plateau method". One of its advantages is that the potential presence of few large aggregates, which can deteriorate the PIA estimate, will be detected by the DFR plateau approach even if all reflectivities in the layer were below the $Z_e$ threshold.

A gradient in DFR can also be caused by, for example, attenuation due to a layer of liquid water. However, in this case, the DFR increases with height, which is in general opposite to the growth of ice particles towards the ground. Hogan et al. (2005) used the DFR gradient in liquid stratocumulus clouds to derive liquid water content profiles. They mention that the DFR profiles must be substantially averaged (they use 1 min and 150 m resolution) before one can exploit the few dBs variation produced by liquid attenuation. Indeed, the DFR profiles can become very noisy due to the random error of reflectivity measurements, especially in case of low signal-to-noise-ratios which can be encountered at cloud tops. Additional noisiness of the DFR signal can be caused by the potential mismatch between multi-frequency antenna beams in regions of strong spatial inhomogeneities. Specifically, another advantage of this technique is that the presence of a DFR Rayleigh plateau is an indication of the good quality of the radar beam alignment. For example, even in a horizontally homogeneous cloud, mispointing could lead to a perceptible DFR gradient. Therefore a rigorous procedure, hinged upon the identification of the DFR plateau at cloud top, is required to derive $\Delta$PIA and assess its quality.

## 3.1 Description of the $\Delta$PIA derivation

The major steps of the DFR Rayleigh plateau method are synthesized in the flow chart of Fig. 2. The approach can be applied to any hydrometeor type, as long as it produces enough differential attenuation for the considered radar frequency pair. In the following, it is exemplified for the $K_a$-W band pair.

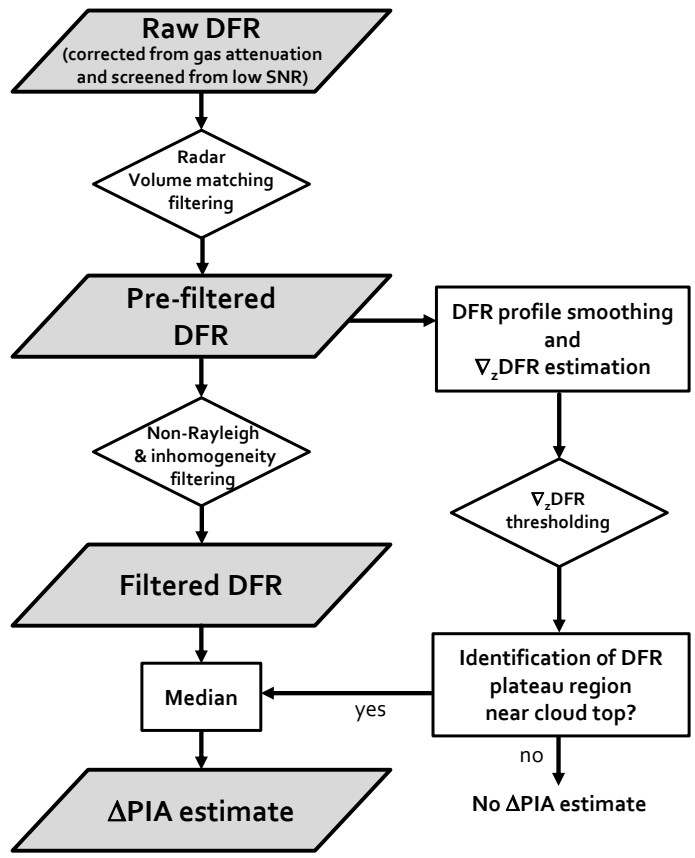

**Figure 2.** Flow chart of the DFR Rayleigh plateau method for estimating the ΔPIA from DFR profiles (see detailed explanations in the text). Parallelogram, diamond and rectangle represent input/output, task and decision, respectively.

The procedure for DFR profiles processing can be divided in two streams. The main stream (left part of Fig. 2) excludes the parts of the profile, which do not fulfill certain quality criteria for deriving ΔPIA. The second stream (right part of Fig. 2) determines the Rayleigh plateau region within the profile. The final ΔPIA is then derived from the filtered DFR which falls into the plateau region.

In the main stream, ΔPIA is derived by taking the median of the DFR profile after it has been successively corrected and filtered according to the following criteria:

**Gas attenuation correction:** Mainly water vapor and oxygen produce non-negligible attenuation, which depends on the radar frequency and must be corrected beforehand. This correction is slightly uncertain (it depends on the quality of the temperature and pressure profiles and on the absorption model used) and may affect the slope of DFR profiles. However, most of the attenuation due to gases is caused by the lowest layers (in our cases, the lowest 2 km), while the Rayleigh plateau region is commonly estimated in high altitude ice clouds. Especially the low-level water vapor profile should

be known well when applying the method to boundary-layer clouds. However, especially for mixed-phase clouds, the liquid-topped cloud structure is more problematic to the method than the uncertainties in the water vapor profile (see also discussion in section 3.3).

**Low Signal to Noise Ratio (SNR):** In order to avoid large errors in the $\Delta$PIA estimate, portions of the profiles contaminated by noise are screened out because the random error in reflectivity increases quickly at low SNR (Hogan et al., 2005). The exact SNR threshold is adjusted for each radar using the two-dimensional frequency distribution of SNR as function of height such as in Tridon et al. (2013b).

**Radar volume matching:** Similarly to Dias Neto et al. (2019), the DFR variance within 20 s by 150 m moving windows must be lower than 4 $\mathrm{dB}^2$ in order to remove the cloud regions potentially affected by a mismatch of the two radar beams;

**Non-Rayleigh scattering and inhomogeneity:** $Z_{m,f_1}$ and its variance (within the same 20 s by 150 m moving windows) must be lower than $Z_{threshold1}$ (5 dBZ at $K_a$-band) and 2.5 $\mathrm{dB}^2$, respectively, in order to exclude the regions where non-Rayleigh scattering is very likely, and where the cloud is highly turbulent;

**Rayleigh plateau detection:** Only the part of the profile, which has been identified as a Rayleigh plateau, is retained.

In the secondary stream, the Rayleigh plateau boundaries are determined from the vertical variations of the DFR. The DFR is first averaged over 20 s by 500 m moving windows; this provides a similar number of averages as in Hogan et al. (2005), except that a finer time resolution is achieved. In order to limit spurious local variations, a polynomial fit of the DFR profile is derived and DFR plateau regions are then defined as portions of the profile for which the absolute value of the DFR gradient is lower than 1 $\mathrm{dB\ km}^{-1}$. Finally, a plateau is confirmed as a Rayleigh plateau only if it has a minimum thickness of 200 m and if it is located less than 500 m from cloud top.

This procedure is then applied to consecutive DFR profiles, and the time evolution of $\Delta$PIA is averaged over a 20 s moving window. If no plateau can be found within this time, no $\Delta$PIA is derived.

## 3.2 Application to example profiles

The methodology is illustrated in Figs. 3 with two profiles taken from the case study presented in Section 5 for which sub-freezing temperatures were recorded at the ground.

The profile in panels (a,b,c) of Fig. 3 depicts two separated cloud layers. The DFR continuously increases in the lower layer (0 to 3 km), while being nearly constant at around 6 dB in the upper cirrus cloud (6 to 8 km). The very low reflectivities (lower than -10 dBZ) indicate that the cirrus is most likely composed of small ice crystals which do not produce any differential scattering signal. This is confirmed by the clear Rayleigh plateau (green shading) in this layer and therefore the DFR in this region can be used for estimating $\Delta$PIA. The fact that the DFR is constant through the whole cirrus cloud indicates that the attenuation must be produced by the lower cloud layer. As will be shown in section 5, the attenuation is caused by a mixed-phase cloud with a total of 6 dB $\Delta$PIA produced by a considerable amount of supercooled liquid water.

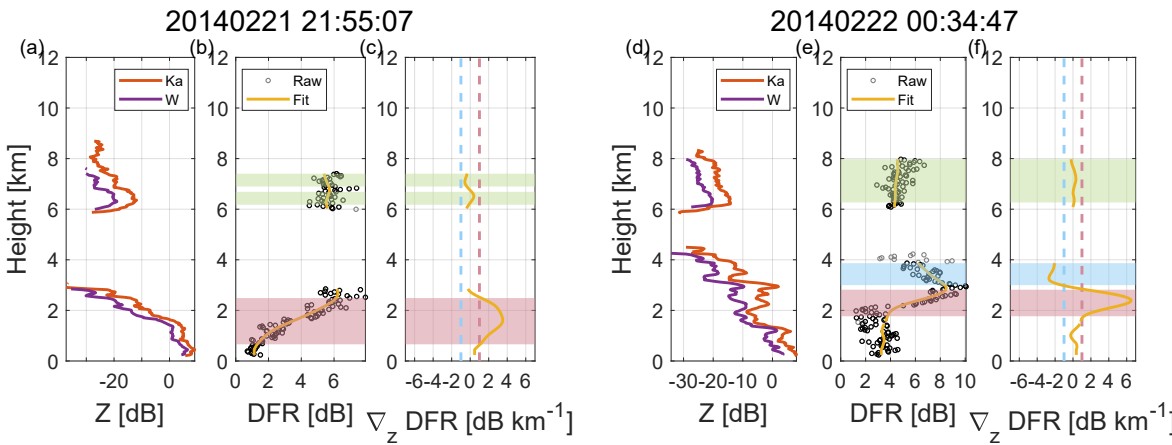

**Figure 3.** Examples of application of the DFR Rayleigh plateau method on two profiles measured at the Hyytiälä Forestry Field site on 21 February 2014 at 21:55:07 UTC (a,b,c) and at 00:34:47 UTC (d,e,f). (a,b) feature the reflectivity measured at $K_a$ and W-band. (b,d) show the corresponding raw (circles) and smoothed (yellow line) DFR, while (c,f) display the DFR gradient (yellow line). Red and blue shadings represent heights where the gradient exceeds $\pm 1$ dB km$^{-1}$ while green shading shows the part of the profile which has been identified as a Rayleigh plateau.

Panels (d,e,f) of Fig. 3 show a profile measured a couple of hours later with the DFR reaching values up to 9 dB before decreasing to about 5 dB. Since the differential attenuation must increase with range, such a decrease can only be explained by a reduction of the differential scattering, indicating that the cloud layer between 2 and 4 km is composed of large snowflakes. Nevertheless, the upper cirrus cloud appears to be still composed of small ice crystals and the Rayleigh plateau identified between 6 and 8 km suggests that a total attenuation of 5 dB is produced by the lower cloud.

### 3.3 General limitations of the method

In some situations, the $\Delta$PIA may not be retrieved because no Rayleigh plateau can be found. For example, the sensitivity of one of the radars may not be sufficient to detect the small particles. In particular, this is likely to happen in case of heavy attenuation due to a thick rain layer, for example. Furthermore, the assumption that hydrometeors grow while they are falling might be violated when a lot of mixing is produced by strong dynamics, for example in case of convective cloud tops or generating cells (Kumjian et al., 2014). Finally, the top of mixed-phased clouds is generally composed of supercooled droplets from which ice particles are formed and grow rapidly. On one hand, the growth of ice crystals might be too quick to produce a 200 m thick Rayleigh plateau. On the other hand, the difference in the dielectric constant of liquid water at $K_a$ and W-band can lead to an overestimation of the DFR by at about 1 dB at 0°C (Lhermitte, 1990) but the presence of few large ice crystals will tend to mitigate this effect. For both reasons, a Rayleigh plateau might be difficult to find for mixed-phased clouds if they are

not topped by an ice cloud. Separating liquid and ice contributions using Doppler spectra could be exploited in these instances (Shupe et al., 2004, 2008; Luke et al., 2010; Kalesse et al., 2016; Li and Moisseev, 2019).

However, in all these situations, the reflectivity-threshold approach would also be erroneous. The Rayleigh plateau method has the advantage that the absence of a Rayleigh plateau is a clear indication that the relative calibration between the radars is troublesome for the corresponding time period.

Even if restrictive criteria are imposed (e.g., minimum thickness and maximum distance from cloud top), a Rayleigh plateau might be erroneously found. In such a case, the height and DFR level of the retrieved plateau are highly variable. Hence, time-continuity criteria can be used to filter out ex post the periods where the algorithm failed.

## 4 Distinct layers of liquid and ice clouds

### 4.1 Case Overview

The first case study was recorded on $20^{th}$ November 2015 at the Jülich Observatory of Cloud Evolution Core Facility (Löhnert et al., 2015, JOYCE) during the TRIple-frequency and Polarimetric radar Experiment for improving process observation of winter precipitation (Dias Neto et al., 2019, TRIPEx). TRIPEx level 2 data are used in this work: the radar data (vertically pointing $K_a$ and W-band) are re-gridded on a common time-height grid and all data have been re-processed and quality controlled (gas attenuation correction and relative calibration) as described in detail by Dias Neto et al. (2019). However, in this latter dataset, the DFR was calibrated using the traditional $Ze$-threshold approach i.e., by determining a relative offset between $K_a$ and W-band reflectivity within 15 min time windows for $Z_{K_a} < -10$ dBZ. Since the current study aims at refining such a procedure, the uncalibrated DFR is first recovered by subtracting this offset from the W-band reflectivity.

A thick ice cloud connected to a cold front was slowly moving over JOYCE on this day with only very weak precipitation (less than 1 mm total accumulation) starting around 11 UTC (Fig. 4a). The DFR reveals growth of larger snow particles starting at temperatures warmer than -15°C (Fig. 4b). In addition to the spatially inhomogeneous DFR structures related to differential scattering, vertical lines of enhanced DFR up to cloud top indicating significant differential attenuation during the precipitating period (e.g. around noon) are visible as well.

### 4.2 Attenuation due to pure liquid cloud

In order to test the DFR Rayleigh plateau method for deriving ΔPIA, it is desirable to find a scenario where the PIA can be clearly attributed to one specific contributor (such as a distinct liquid cloud layer). Such a situation seems to be present during the morning hours (4-5 UTC), when some low level, non-precipitating clouds are visible in the radar reflectivity time-height plot (indicated by the first square in Fig. 4a).

When zooming into this period (Fig. 5a), two main layers of cumuliform clouds clearly appear to be distinct from the upper ice cloud. The banded DFR signature visible in the upper ice cloud seems to be related to the presence of these shallow clouds. Excluding the parts of the cloud which clearly contain non-Rayleigh scattering particles, there isn't any perceptible DFR

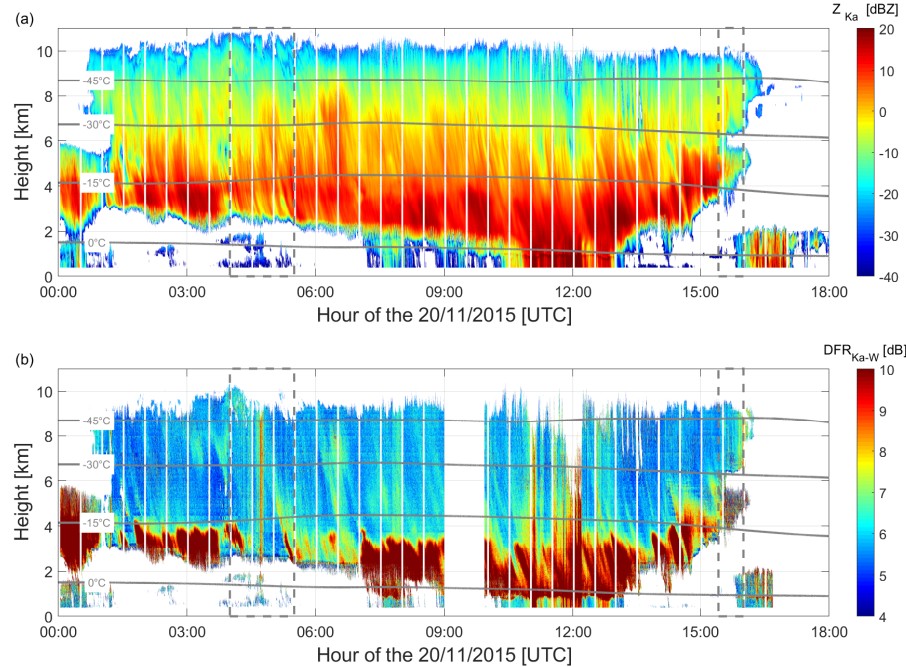

**Figure 4.** Time-height [UTC-km] plots of the (a) $K_a$ reflectivity and (b) $K_a$-W DFR observed at the Jülich Observatory on 20 May 2014. The black lines show the -45, -30, -15 and 0°C isotherms.

increase in the ice cloud, which indicates that the majority of the attenuation signal is caused by the lower level clouds. The co-located ceilometer features typical strong backscattering signals (not shown) at cloud bottom combined with full extinction of the lidar signal above, which is a typical signature of the base of liquid clouds (magenta points in Fig. 5a). Even though the top of some of the cumuliform clouds reaches temperature slightly below 0°C, the clouds are assumed to be purely liquid. This is also confirmed by the slight increase of reflectivity with height, which is expected for an adiabatically increasing liquid water content. Not only an opposite reflectivity gradient would be expected for ice containing or mixed-phase clouds, but also the reflectivity of droplets would dramatically decrease while freezing because of the smaller refractive index of ice.

If these assumptions are correct and the attenuation signal is mainly caused by the lower level clouds, there should be a high correlation between the derived ΔPIA and the liquid water path (LWP) derived from a co-located microwave radiometer. According to the processing steps described in Section 3, the filtered DFR depicted in Fig. 2 is obtained by screening out areas which indicate problematic radar volume matching (for example, close to cloud edges) or non-Rayleigh scattering particles (highlighted by gray shading in Fig. 5b). As indicated by the black contour line in Fig. 5b, the Rayleigh plateau method is able to make use of a much thicker part of the ice cloud (up to 5 km) for estimating ΔPIA as compared to the $Z_e$ threshold method ($Z_{K_a} < -10$ dBZ). The threshold method would only use the uppermost 1 km of the ice cloud, which includes cloud areas that are prone to volume matching issues and increasingly affected by the different sensitivity limits of the radars.

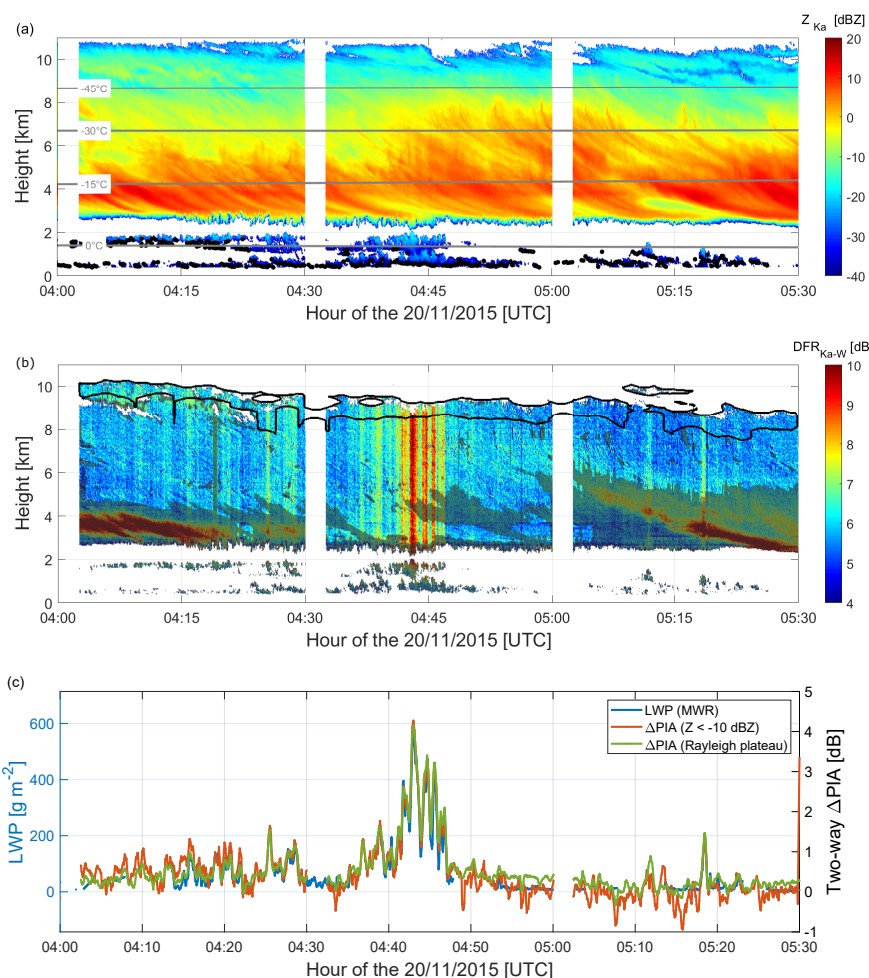

**Figure 5.** Zoom into the period of low level liquid clouds from the case study shown in Fig. 4 (earliest square). Time-height [UTC-km] plots of (a) $K_a$ reflectivity with liquid cloud base detected by the co-located ceilometer as magenta points and (b) $K_a$-W DFR. Parts which have been detected by the filtering criteria to contain non-Rayleigh scattering particles are highlighted by the gray shading. (c) Time series of the $K_a$-W DFR at cloud top, i.e. the two-way $\Delta$PIA derived with the $Ze$-threshold approach and with the Rayleigh plateau method (scale on the right y-axis) overlaid by the LWP measured by the MWR (scale on the left y-axis). Note that the scaling of the $\Delta$PIA axis was adjusted to match LWP at 0°C, i.e. assuming a two-way liquid water attenuation of 7 dB g$^{-1}$ m$^2$. The black lines show the -45, -30, -15 and 0°C isotherms in (a) and the uppermost $Z_{K_a} = -10$ dBZ contour in (b).

The reflectivities of the two radars are adjusted so that the $\Delta$PIA is equal to zero when the ice reflectivity remains small throughout the profile and when the LWP obtained from the co-located Humidity and Temperature PROfiler (HATPRO, Rose et al. (2005)) is negligible (i.e. between 01:00 and 02:00 UTC). The LWP is derived from the seven channels along the 22 GHz water vapor absorption band using a statistical retrieval similar to Löhnert and Crewell (2003). The $\Delta$PIAs derived from the Rayleigh plateau method and the $Ze$-threshold approach are shown in Fig. 5c together with the LWP. Data gaps in LWP retrievals are due to regular MWR calibration procedures and intermediate azimuth/elevation scans. Note that slightly negative LWP values are expected to occur close to the detection limit of the MWR due to the statistical retrieval applied. Strong and sharp LWP variations are found at the scale of less than a minute. They are clearly correlated with the presence of high reflectivity low-level cumulus clouds (auxilliary measurements from the ceilometer before and after the low cumulus clouds confirm that the upper cloud is composed of ice only) and with the $\Delta$PIA variations. In order to avoid discrepancies due to too long time averaging, a relatively high time resolution of 20 s is chosen for the retrieval of $\Delta$PIA (Section 3) in order to account for the fast LWP variations in the observed cumuliform clouds.

The agreement between the time series of MWR derived LWP and $\Delta$PIA (Fig. 5c) is remarkable for both methods. Overall, the retrieval with the Rayleigh plateau method appears less noisy than with the $Z_e$ threshold approach. In particular, nonphysical negative values found by the $Z_e$ threshold approach between 05:00 and 05:30 UTC are probably due to the reliance of a few measurements at cloud top, where the SNR is low and the random error in reflectivity is large. During short time periods (04:50 and 05:00 UTC), the $Z_e$ threshold approach appears to perform better than the Rayleigh plateau method. We speculate that this is due to a region of slightly enhanced DWR which has very small vertical gradient. However, overall the Rayleigh plateau method provides more consistent results.

Fig. 6a shows the scatter plot of LWP and $\Delta$PIA derived with the Rayleigh plateau method for the full case, with the points from cumuliform clouds during the morning hours shown in Fig. 5 highlighted with a different color scale. The data follow closely the relation predicted by liquid water refractive index models for a temperature of 0°C, i.e. about 7 dB two-way $K_a$-W differential attenuation per $\mathrm{kg\ m^{-2}}$ of LWP. By selecting one of the refractive index models (e.g., Turner et al., 2016), LWP is retrieved from $\Delta$PIA and the performances of both methods are compared statistically in Fig. 6b for the zoomed period of Fig. 5. While the obtained slight positive biases are similar for both methods and can be easily explained by the unaccounted attenuation produced by the thick ice cloud, the Rayleigh plateau method seems to outperform the reflectivity threshold approach in terms of standard deviation.

These results suggest that LWP larger than roughly $100\ \mathrm{g\ m^{-2}}$ can safely be retrieved with the $\Delta$PIA method provided that the PIA is due to liquid cloud water only. In Fig. 6a, no point can be shown during rainy periods (before 01:00, between 07:00 and 08:00, 9:00 and 13:00 and after 16:00 UTC) because the presence of rain drops violates the MWR retrieval assumptions (scattering effects are assumed to be negligible). Nevertheless, the retrieved $\Delta$PIA during these periods might be useful because it can provide information on the integrated amount of rain and help to constrain radar retrievals, given that the effect of wet radome attenuation can be mitigated (as it was the case for the W-band radar used during TRIPEx, which was equipped with strong blowers Küchler et al., 2017).

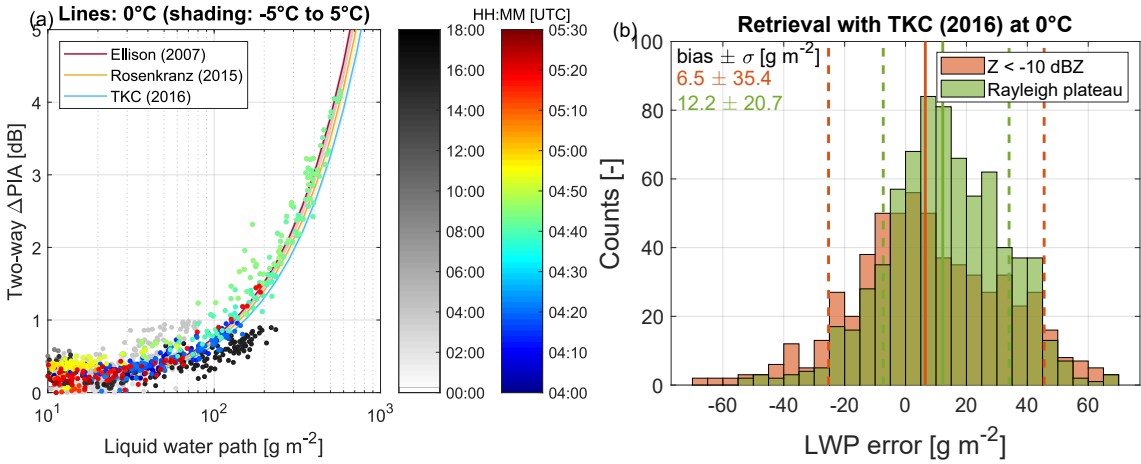

**Figure 6.** (a) Scatterplot of two-way ΔPIA retrieved using the Rayleigh plateau technique as function of the LWP measured by the MWR for the entire case study (color of the circles denotes the time in UTC). (b) Statistics (probability density function, bias and standard deviation) of the retrieved LWP for the zoomed period of Fig. 5 using the Turner et al. (2016) model at 0 °C for the traditional reflectivity threshold and the Rayleigh plateau method using the MWR as a reference.

### 4.3 Observed temperature dependence of differential attenuation due to liquid water

The close relation between ΔPIA and LWP shown in Fig. 6a seems to significantly change for the later period between 15:30 and 16:00 UTC (black points). The case overview (Fig. 4) shows that the low-level liquid clouds have mostly disappeared but the thick ice cloud has separated into two distinct layers with the upper one between 7 and 10 km and the lower one between 4 and 6 km. Interestingly, while the DFR structure of the upper layer appears to be similar to the earlier periods, the DFR in the lower ice cloud is extremely noisy. By experience with other instances, it is suspected that the radar beam mismatch

is amplified in this cloud layer due to high spatio-temporal variability inside this cloud (for example caused by wind shear, convection, or turbulence).

A zoomed view of this time period (Fig. 7) reveals indeed a very high variability of reflectivity and DFR in time and space in the ice cloud between 4 and 6 km. The signatures appear to be similar to generating cells (Kumjian et al., 2014), which are often observed as a result of instabilities at the top of ice clouds. When looking more carefully, a high correlation is revealed between

the columns of enhanced reflectivity and both ΔPIA and LWP. As suggested by the co-located ceilometer (liquid cloud base detected around 5 km in Fig. 7a), supercooled liquid is generated by updraughts in this columns even though the temperature level is ≈-25°C. This lower temperature also explains why the ΔPIA and LWP lines are not matching quantitatively in Fig. 7c (where the y-axis are set to match at 0 °C) and the different cluster of ΔPIA and LWP for this period in Fig. 6. At colder temperatures, the $K_a$-W differential attenuation produced by liquid droplets is simply much smaller for the same LWP (see

Section 2). When plotting the model prediction for ΔPIA and LWP for the lower temperature range (Fig. 8a), the retrieved LWP and ΔPIA are fairly consistent with the liquid water refractive index models (the apparent overestimation of ΔPIA by

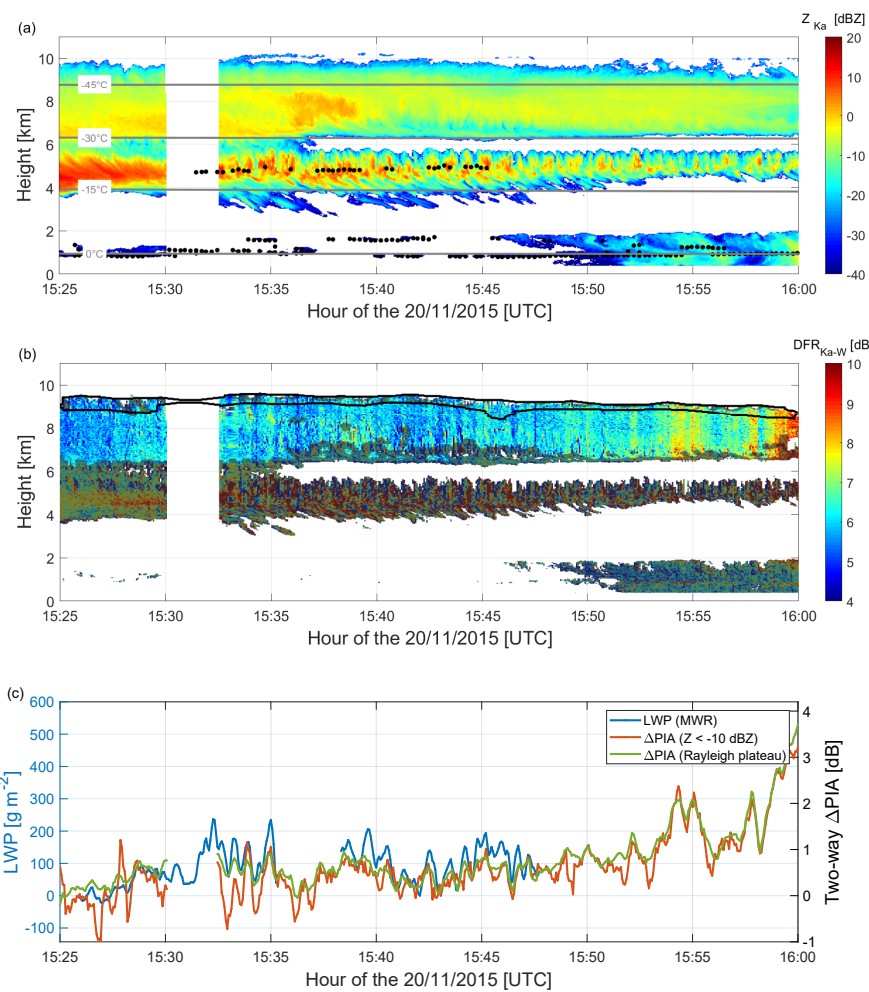

**Figure 7.** Same as Figure 5 but for the period between 15:25 and 16:00 UTC where a second liquid layer is detected by the ceilometer at around 5 km, i.e. at temperatures comprised between -15 °C and -30 °C. Note that the adjustment of y-axis scales follow the same convention as Figure 5 (i.e. ΔPIA and LWP matching at 0 °C). For the colder temperature of this cloud, the same LWP is expected to produce less attenuation than at 0 °C, as the curves in (c) suggest.

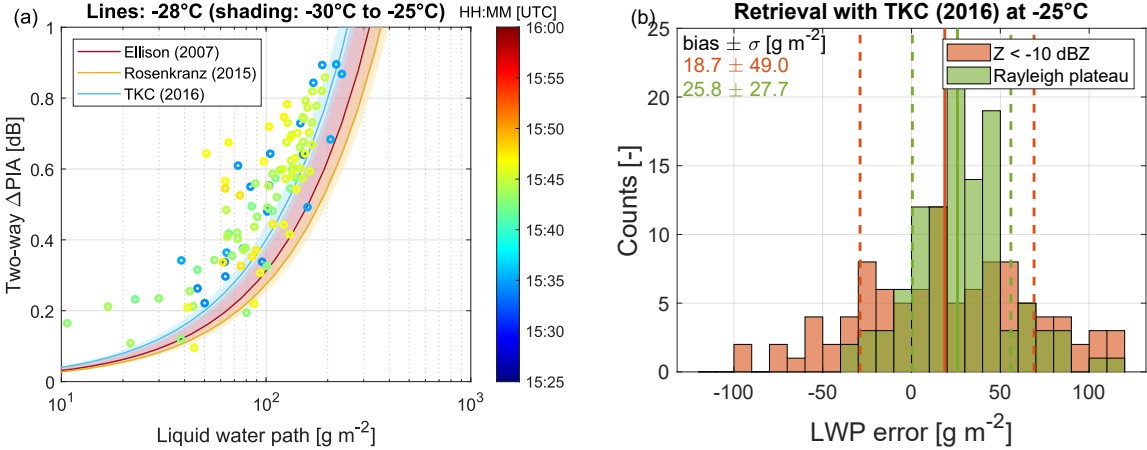

**Figure 8.** Same as Fig. 6 but for the period between 15:25 and 16:00 UTC.

about 0.2 dB can again be easily explained by ice attenuation). Because of the lower slope of the theoretical curves at colder temperatures, the bias in retrieved LWP (Fig. 8b) appears to be slightly larger compared to the earlier period. However, the retrieval using the Rayleigh plateau method shows again a significantly lower standard deviation than the reflectivity threshold
approach.

Under ideal conditions, such co-located MWR and $K_a$-W radar observations would allow a determination of the mean liquid water temperature. A similar rationale has been presented by Matrosov and Turner (2018), where only passive microwave observations in the $K_a$ and W-bands have been used. The radar approach provides additional information about the profile, which a pure MWR method is unable to capture. Regarding a potential profiling of liquid water inside e.g. mixed-phase clouds,
it is worth noting that the temperature sensitivity of ΔPIA is much stronger when using radar channels in the G-band (note the steep decrease of the purple lines at sub-freezing temperatures in Fig. 1). Therefore, the recent development of G-Band radar technology (Cooper et al., 2018) may unlock the potential of such systems for profiling liquid water when combined with low-frequency MWR.

With this case study example, the objective was mainly to test how reliably and consistently the ΔPIA can be retrieved by
the Rayleigh plateau method. Although situations with ice cloud above shallow liquid layers might not be uncommon at many sites, the origin of the ΔPIA signal is in general more complex and due to various sources, which will be investigated in the following section.

## 5   Mixed-phase clouds and intense snowfall

The Rayleigh plateau method is now tested on a more complex case which comprises mixed ice and supercooled liquid as well
as snowfall on the ground. This case is characterized by a frontal passage that occurred on 21st February 2014 at the Hyytiälä

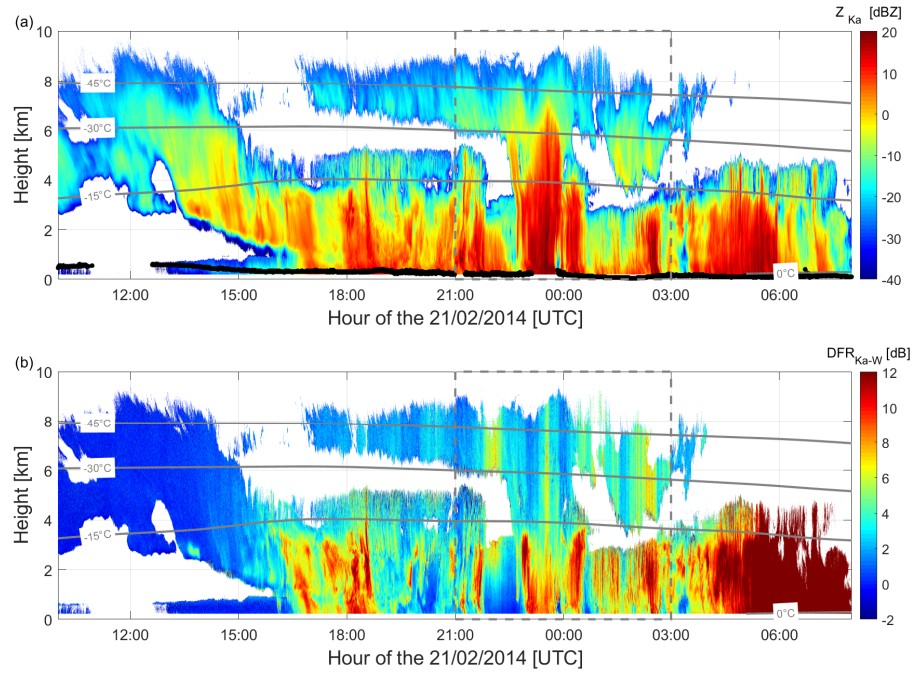

**Figure 9.** Time-height [UTC-km] plots of the (a) $K_a$ reflectivity with liquid cloud base detected by the co-located lidar as magenta points and (b) $K_a$-W DFR observed at the Hyytiälä Forestry Field site on 21 February 2014. The black line shows the -45, -30, -15 and 0°C isotherms.

field site, Finland (Fig. 9). The multi-frequency radar and auxiliary dataset have been recorded in the framework of the Biogenic Aerosols Effects on Clouds and Climate (BAECC) field experiment (Petäjä et al., 2016), during which the U.S. Department of Energy Atmospheric Radiation Measurement (ARM) deployed the second ARM mobile facility (AMF2). Ground-based in-situ and multi-frequency radar observations of this particular case have been extensively studied and described in Kneifel
et al. (2015), Kalesse et al. (2016), Moisseev et al. (2017), von Lerber et al. (2017), Mason et al. (2018) and Kneifel et al. (2020).

In this work, reflectivity data from the $K_a$-band ARM zenith radar (KAZR; Isom et al. (2014b), Fig. 9a) and the marine W-band ARM cloud radar (MWACR; Isom et al. (2014a)) have been corrected for gas attenuation and re-gridded on a common time-height grid and cross-calibrated after compensation of mismatches in time and range. Indeed, using the cross-correlation
of the reflectivity fields, it was found that best matching was obtained using offsets in range (30 m, i.e., one range gate) and time (0 to 4 s between 0 and 5 km and 4 s above). The KAZR highest sensitivity (KAZRMD) and the general (KAZRGE) modes are properly inter-calibrated and then merged in order to maximise the SNR and avoid receiver saturation close to the ground (below approximately 1 km). In Fig. 9b is depicted the DFR corresponding to SNR larger than -16 and -17.5 dB for KAZR and MWACR, respectively.

This case represents a typical mixed-phase cloud, with persistent supercooled liquid layers as shown by the liquid cloud based detected by the co-located lidar (magenta points in Fig. 9a). These supercooled liquid clouds extend up in the atmosphere as suggested by radiosoundings of 17:00 and 23:00 UTC. Relative humidity with respect to liquid water was close to saturation up to 5 km and 3 km altitude, respectively (not shown). This leads to significantly rimed snow and graupel at the ground, as confirmed by the large bulk particle density (comprised between 200 and 500 kg m$^{-3}$) retrieved from in-situ (Moisseev et al., 2017; von Lerber et al., 2017) and multi-frequency radar retrieval (Mason et al., 2018). Conversely, the period with stark reflectivities of more than 10 dBZ up to 6 km altitude just before midnight (Fig. 10a) was found to be dominated by large aggregates and snow rate up to 4 mm h$^{-1}$. A detailed analysis of this time period (Kalesse et al., 2016) shows that the liquid topped mixed-phase cloud starts being seeded around 23:00 UTC by ice falling from the upper cirrus cloud. During this time period, the lower lidar backscatter (see Fig. 4c in Moisseev et al. (2017)) reveals that intense seeder-feeder process depletes supercooled water (note the gap in the detected liquid cloud base between 23:10 and 23:50 UTC in Fig. 9a), which leads to a reduction of riming.

As in the TRIPEX case, differential scattering due to large snowflakes is obvious in the spatially inhomogeneous DFR structures in the lower cloud layer (below 4 km in Fig. 9b). Furthermore, periods of significant differential attenuation can be identified as vertical lines of enhanced DFR in the cirrus cloud (e.g. around 22:00, 23:30 and 02:30 UTC), which is expected to be composed of small ice crystals only. Again, the $\Delta$PIA has been derived from the DFR Rayleigh plateau method and the following analysis will focus on this specific time period (dashed-line square in Fig. 9). Data after 03:00 UTC show high DFR (larger than 12 dB) due to the considerable non-Rayleigh scattering and attenuation produced by rain and melting ice (and possibly, by radome attenuation). Though this information could be very useful in a full-column precipitation retrieval, profiles after 03:00 UTC are not further considered in this analysis because of the complexity of attenuation sources.

## 5.1 Path-integrated attenuation due to liquid cloud and snow

In the previous case, the lower level liquid cloud could be identified to be the major contributor to the attenuation signal. In this case, however, the possibility of attenuation due to ice and snow must also be considered as discussed in Section 2. A potential snow and ice attenuation signal should be detectable by comparing $\Delta$PIA to the LWP measured by co-located MWR, which is insensitive to snow scattering at the low frequency bands used.

Similarly to the TRIPEX case, areas with high inhomogeneity and non-Rayleigh scattering are successfully filtered out by the DFR Rayleigh plateau method (gray shading in Fig. 10b). In this case, the $Z_{K_a} < -10$ dBZ threshold retains a slightly smaller extent of data for deriving $\Delta$PIA. Because of the presence of the upper cirrus cloud during most of this case study, misclassification of Rayleigh targets by the $Z_e$-threshold approach are rare. Nevertheless, when the two cloud layers are connected or when the upper layer cloud is absent (e.g., just after 00:00 and just before 01:00 UTC, respectively), $Z_{K_a}$ at the top of the lower cloud layer often satisfies the -10 dBZ threshold while the corresponding large DFR has clearly a non negligible contribution from non-Rayleigh scattering (large) particles. This illustrates again the benefit of using the Rayleigh plateau method instead of a fixed $Z_e$ threshold.

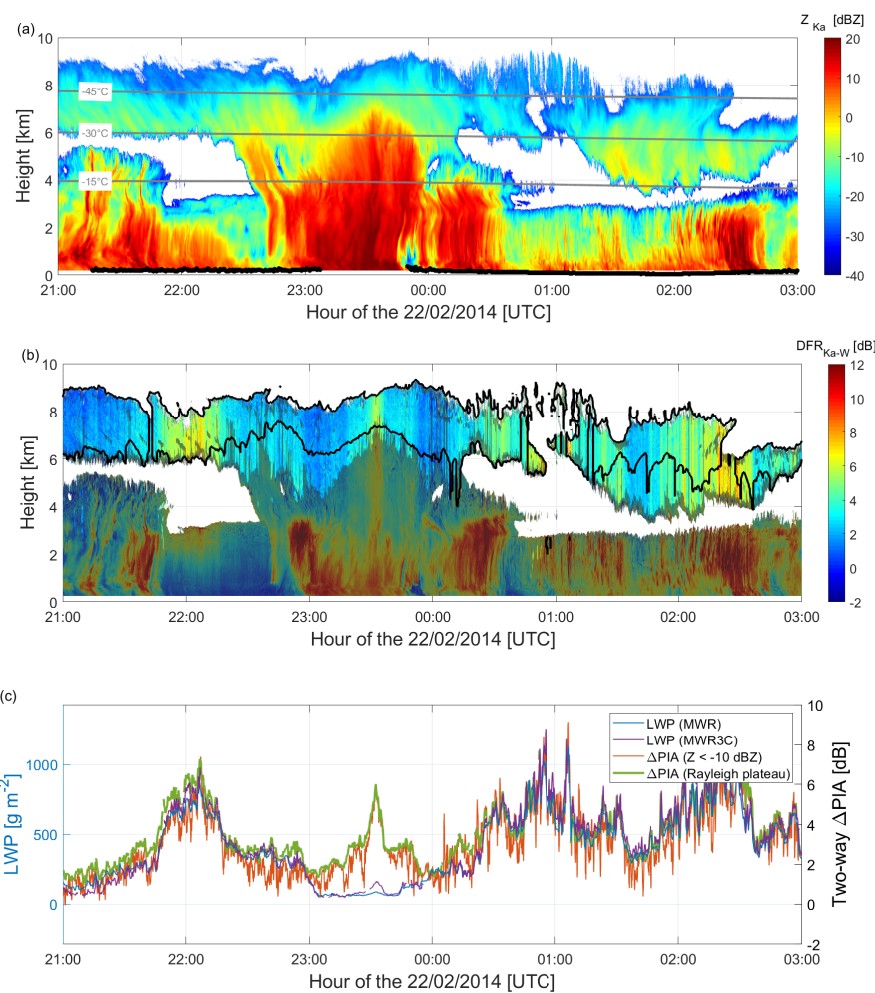

**Figure 10.** Zoom into the periods of high LWP or intense snowfall. Time-height [UTC-km] plots of the (a) $K_a$ reflectivity with liquid cloud base detected by the co-located lidar as magenta points and (b) $K_a$-W DFR filtered for non-Rayleigh targets (gray shading). (c) Time series of the $K_a$-W DFR at cloud top, i.e. the two-way $\Delta$PIA derived with the $Z_e$-threshold approach and with the Rayleigh plateau method (scale on the right y-axis) overlaid by the LWP measured by the MWRs (scale on the left y-axis). The black lines show the -45, -30 and -15°C isotherms in (a) and the uppermost $Z_{K_a} = -10$ dBZ contour in (b).

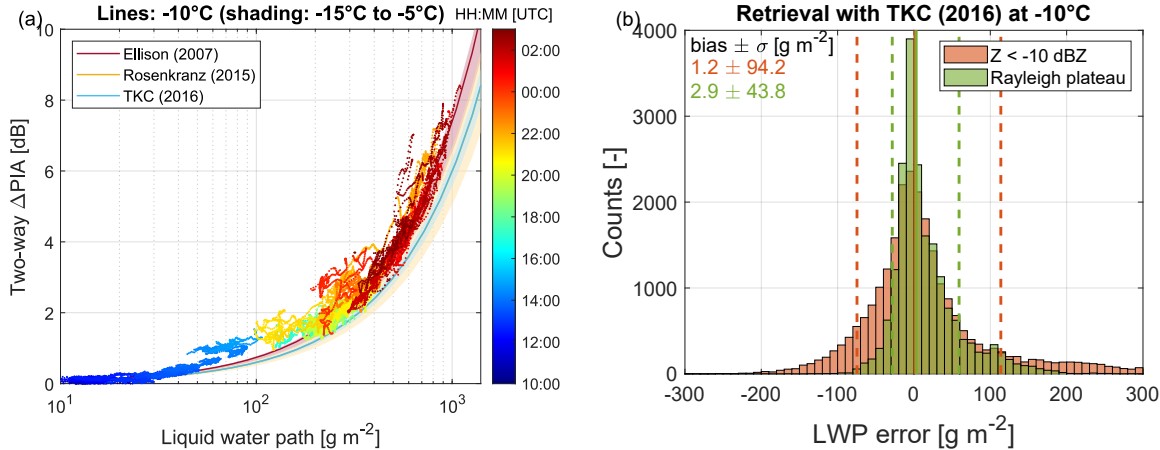

**Figure 11.** Scatterplot of the two-way $\Delta$PIA retrieved using the Rayleigh plateau technique as function of the LWP measured by the two-channel MWR (color of the points denotes time in UTC). (b) Statistics (probability density function, bias and standard deviation) of the retrieved LWP for the corresponding time period using the Turner et al. (2016) model at -10°C for the traditional reflectivity threshold and the Rayleigh plateau method using the two-channel MWR as a reference.

In Fig. 10c, $\Delta$PIA obtained from the Rayleigh plateau method and the $Z_e$-threshold approach are compared to the LWP measured by the co-located ARM microwave two-channel (MWR, 23.8 and 31.4 GHz, Cadeddu and Ghate (2014b)) and

three-channel (MWR3C, 23.8 and 30 and 89 GHz, Cadeddu and Ghate (2014a)) radiometers. Except for the high reflectivity period between 23:00 and 00:00 UTC, a fairly good agreement is found between the $\Delta$PIA timeseries of both methods and LWP suggesting that most of the differential attenuation is produced by cloud liquid water. Again, strong variations on timescales of less than a minute can be found in the LWP and, consequently, on the DFR timeseries (e.g. around 01:00 UTC). Like for the TRIPEX case, the retrieval from the $Z_e$-threshold approach appears noisier than the Rayleigh plateau method. Furthermore,

mis-classification of Rayleigh targets by the $Z_e$-threshold approach leads to erroneous spikes of $\Delta$PIA which would appear more frequent without the upper cirrus cloud.

In Fig. 11a, the retrieved $\Delta$PIA (with the 23:00 to 00:00 UTC period filtered out) are compared to the LWP measured by the two-channel MWR. The relations predicted by liquid water refractive index models for temperatures between -15 and -5°C are used as a reference. These temperatures correspond to the heights where the supercooled liquid water clouds are

expected according to radiosoundings. The retrieved $\Delta$PIA appears to be slightly too large in comparison to the two-channel LWP. This small overestimation is particularly visible when reflectivity is high (e.g., see Fig. 10 around 21:30, 00:20 and 02:30 UTC) and may be due to snow attenuation, as shown in the next section. Note that for this temperature range, the liquid water refractive index models slightly disagree: at -10°C, the Ellison (2007) model predicts 7 dB two-way $K_a$-W differential attenuation per kg m$^{-2}$ of liquid water while both Rosenkranz (2015) and Turner et al. (2016) models only predict 6 dB.

Although the Turner et al. (2016) model was specifically developed for temperatures as low as -32°C, the radar data seem to

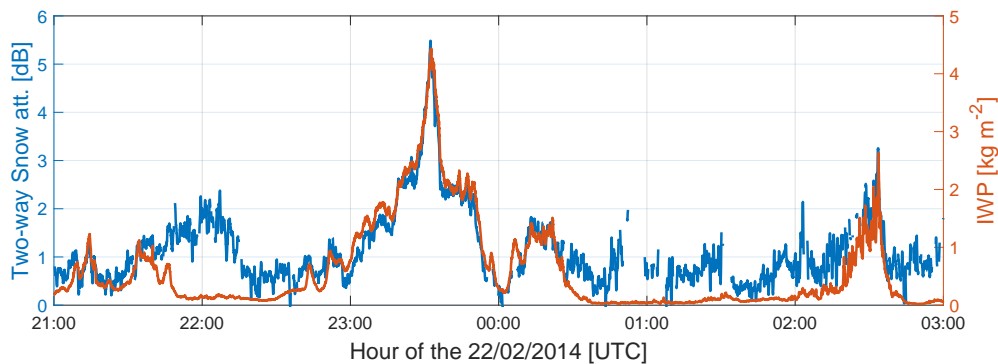

**Figure 12.** Two-way snow attenuation (scale on the left y-axis) against IWP (scale on the right y-axis) as function of time in UTC. Snow attenuation was derived from the difference between observed $\Delta$PIA and calculated liquid water attenuation based on LWP retrieved by the MWR. IWP was derived with the IWC-$Z_{K_a}$ relation proposed by Protat et al. (2007).

agree better with the Ellison (2007) model. However, no definitive conclusions can be drawn about refractive index models at sub-freezing temperatures because of the presence of not well-quantified additional sources of attenuation (e.g. from snow). The Turner et al. (2016) model is chosen for computing LWP from the retrieved $\Delta$PIA and for comparing the performances of the two methods in Fig. 11b (where the 23:00 to 00:00 UTC period has again been filtered out). As expected, LWP error

reaches larger values than in Fig. 6b because higher LWP is observed on average in the BAECC case. Similar to the TRIPEX case, the Rayleigh plateau method outperforms the $Z_e$-threshold method in terms of standard deviation. Furthermore, in the error distribution of the Rayleigh plateau method, the apparent skewness toward large values can be explained by some snow attenuation in a minority of profiles.

### 5.2 Path-integrated differential attenuation due to snow

Attenuation due to snow and ice depends mainly on the total ice mass in the column but larger sizes and higher degree of riming are expected to further enhance the attenuation signal for a given snow mass (Battaglia et al., 2020a). This effect can be seen during the high reflectivity period (between 23:00 and 0:00 UTC), where $\Delta$PIA is found to reach up to 6 dB, while LWP remains low (Fig. 10c). $\Delta$PIA variations during this time period seem correlated with the radar reflectivity field, suggesting that the differential attenuation is produced by the intense snow composed of large aggregates (snow rate of 4 mm h$^{-1}$ and

median mass diameters up to 5 mm, as retrieved from co-located ground-based in-situ instruments by Moisseev et al. (2017)).

In order to separate $\Delta$PIA due to liquid and snow, the contribution of the MWR-retrieved LWP to $\Delta$PIA is calculated using the refractive index model from Turner et al. (2016) assuming a temperature of -10°C. The difference to the total measured $\Delta$PIA can then be assigned to snow attenuation. As a consistency check, the ice water content (IWC) is derived from $Z_{K_a}$ in dBZ using the relation proposed by Protat et al. (2007) for mid-latitudes ($\log_{10}(\text{IWC}) = 0.000372 Z_{K_a} T + 0.0782 Z_{K_a} -$

0.0153$T$ − 1.54 with $T$ the temperature in °C, and integrated over the altitude in order to obtain the ice water path (IWP). The resulting timeseries of snow attenuation and IWP are generally well correlated (Fig. 12). In particular, the peak of large snow attenuation deduced from the mismatch between LWP and $\Delta$PIA can be well explained by the corresponding large IWP. The presence of large snowflakes at 23:30 UTC is also supported by the disagreement between the MWR and MWR3C retrievals in Fig. 10c: while brightness temperatures at 30 GHz and below are relatively flat around this time, a slight enhancement of 6 K is observed at 89 GHz (not shown). Such a behaviour is consistent with an enhanced scattering produced by large snowflakes (Kneifel et al., 2010). The snow and ice present in this case produced roughly 1.2 dB two-way attenuation per kg m$^{-2}$. These values are in agreement with self-similar Rayleigh-Gans attenuation computations for low-density aggregates of characteristic size equal to 5 mm (Fig. A6 in Battaglia et al. (2020a)). While this seems lower than the attenuation measured by Nemarich et al. (1988) using a horizontal link, this is larger than what is obtained when using the more recent relations found by Matrosov (2007) ($k_{e,W} = 0.12S^{1.1}$ where $S$ is the snow rate in mm h$^{-1}$) and Protat et al. (2019) ($2k_{e,W} = 0.0413Z_W$).

Finally, for short time periods in Fig. 12, the disagreement between the total $\Delta$PIA and liquid attenuation suggests significant snow attenuation estimate while the IWP remains very low (e.g. around 22:00 and after 0:30 UTC). Interestingly, this corresponds to periods where very high LWPs were measured by MWRs (LWP > 0.6 kg m$^{-2}$ in Fig. 10c). Such high LWPs are at the edge of the range of applicability of MWR retrievals. In fact, these conditions are likely favorable for the formation of drizzle drops which lead to larger differential attenuation per unit mass (section 2), thus violating the MWR retrieval assumptions. High LWPs are also conducive for riming and heavily rimed snowflakes, which are very efficient in producing attenuation at W-band (Battaglia et al., 2020a). This second assumption is supported by the extensive characterization of snow properties for this case study: both ground-based in-situ measurements (Moisseev et al., 2017; von Lerber et al., 2017) and triple-frequency radar retrievals (Mason et al., 2018) suggest a particularly large bulk density (more than 300 kg m$^{-3}$) around 22:00 UTC and between 00:30 and 03:00 UTC, except for a short time period around 02:30 (see Fig. 10d in Mason et al., 2018). Both circumstances can explain the disagreement between observed and retrieved $\Delta$PIA and LWP.

## 6   Conclusions

Multi-frequency radar retrievals often require as important integral constraint a reliable estimate of the differential path integrated attenuation ($\Delta$PIA) caused by gases, rain, melting hydrometeors, cloud liquid water, and snow. While $\Delta$PIA can be relatively easy derived from nadir pointing radars, it is common practice for ground-based radars to derive $\Delta$PIA by matching reflectivities at cloud top, where only Rayleigh targets with identical effective reflectivities ($Z_e$) are expected.

While this method works in many situations, it also has inherent problems: low concentrations of medium sized (non-Rayleigh) particles might produce $Z_e$ below the threshold, but they would be inappropriate for the $Z_e$ matching. Conversely, a high concentration of small particles could exceed the threshold, while still being valid Rayleigh-scattering particles. Finally, the transition from Rayleigh to non-Rayleigh scatterers does not only depend on particle size but also on radar frequency which makes definition of $Z_e$ thresholds even more ambiguous. In the new approach presented in this work, the aim was instead to identify signatures that particles in the vicinity of cloud top are small enough to scatter in the Rayleigh regime. For this, the key

approach is searching for a Rayleigh plateau, i.e. a large enough region where the vertical gradient of DFR remains below a certain threshold. With this method, the region from which the $\Delta$PIA can be derived is usually substantially enlarged compared to the $Z_e$ threshold method. It also provides an indication of the quality of the DFR (e.g., appropriate beam alignment), which is usually not addressed by the $Z_e$ threshold method.

The new methodology is applied to two mid-latitude case studies representing different complexity of clouds. With a distinct low-level liquid cloud and a thick ice cloud aloft, the first case represents an ideal scenario to test the method, because the attenuation can be solely attributed to the lower liquid cloud. The comparison of the derived $\Delta$PIA with time series of LWP from co-located MWR shows a remarkable agreement, sometimes even for LWP values lower than $100\,\mathrm{g\,m^{-2}}$. The second case represents a more commonly observed complex cloud occurrence including mixed-phase clouds and intense snowfall. Also in this case, from the comparison with the MWR, the $\Delta$PIA is predominantly produced by cloud liquid. In the absence of a MWR, the $\Delta$PIA method appears to be an alternative approach to retrieve LWP in situations where other sources of attenuation can be assumed to be negligible. If a co-located low frequency ($f < 90\,\mathrm{GHz}$) MWR is available, the $\Delta$PIA technique can also be used to infer the approximate height of the liquid layer due to dependence of the liquid attenuation on temperature. During snowfall events or thick ice clouds without a melting layer or rain, the comparison of $\Delta$PIA and LWP can be used to infer the attenuation signal caused by the frozen hydrometeors in the column. In the case study analyzed, the $\Delta$PIA is generally very consistent with a radar derived ice water path. Noticeably, a deep snow system produces as much as $5\,\mathrm{dB}$ $\mathrm{K}_a$-W differential attenuation, which corresponds approximately to a snow attenuation coefficient of $1.2\,\mathrm{dB}$ attenuation per $\mathrm{kg\,m^{-2}}$ for this specific event. Such values are within the range of the few available relations in the literature. However, the differences between the previously published relations might indicate a large dependence on the properties of snow particles (e.g. size, rimed mass fraction), which needs to be investigated on a larger data set.

In order to quantitatively assess the improvement brought by the new Rayleigh plateau method, LWP has been retrieved from $\Delta$PIA estimated by both methods using identical liquid water refractive index model. For the two cases analyzed, the new methodology shows a much smaller spread in the differences to the reference MWR retrieved LWP. In addition, the main assets of the new method are: 1) it can be applied independently of the radar frequency pair (without the need of fine tuning a $Z_e$ threshold). The method exploits 2) a much a larger region of the cloud to derive $\Delta$PIA (which should lead to a better accuracy in general). Finally, it provides 3) quality controlled estimates (no $\Delta$PIA can be retrieved if no Rayleigh plateau is found) while the $Z_e$ threshold approach can lead to erroneous estimates.

In future work, this procedure will be systematically applied to a growing data set of $\mathrm{K}_a$-W band radar and MWR observations in order to thoroughly characterize snow attenuation at W-band, a key parameter for the retrieval of snow properties from space-borne radars and MWRs. Quality-controlled smooth DFR profiles, a by-product of the technique, could also help to improve microphysical process studies. In principle, this technique can even be extended to scanning multi-frequency radars (such as the scanning ARM cloud radars) where the liquid and snow attenuation signals would be enhanced due to the longer path lengths.

In order to further disentangle the differential attenuation and scattering signal, the analysis of the multi-frequency Doppler spectra will be necessary. While several studies looking at rain and melting layer made significant progress in this direction,

they also found that the quality and in particularly, the requirement on radar volume matching, are very high. The incorporation of Doppler spectra in combination with newly developed G-Band radars is expected to bear great potential for profiling liquid water and snow even within thick mixed-phase clouds.

*Data availability.* BAECC data were obtained from the U.S. DOE ARM Climate Research Facility www.archive.arm.gov (Cadeddu and Ghate, 2014a, b; Isom et al., 2014a, b). TRIPEX radar data was made available by Dias Neto et al. (2019) on the ZENODO platform (https://doi.org/10.5281/zenodo.1341390).

*Author contributions.* Data analysis and implementation of the DFR Rayleigh plateau method was made by FT. Conceptualization of the method, interpretation and writing was shared between FT, AB and SK.

*Competing interests.* The authors declare that they have no conflict of interest.

*Acknowledgements.* This work was part of the project "Ice processes in Antarctica: Identification via multiwavelength active and passive measurements and model evaluation" (DE-SC0017967) funded by the Atmospheric System Research. Contributions from S. Kneifel have been funded by the Deutsche Forschungsgemeinschaft (DFG, German Research Foundation) under grant KN 1112/2-1 as part of the Emmy-Noether Group OPTIMIce. The authors wish to thank Bernhard Pospichal, Jan Schween and Maria Cadeddu for useful discussions about the IGMK (Institute for Geophysics and Meteorology at the University of Cologne) microwave radiometer, the IGMK ceilometer and ARM microwave radiometers, respectively. TRIPEx data were collected at the JOYCE Core Facility (JOYCE-CF) co-funded by the German Research Foundation research grant LO 901/7-1. Major instrumentation at the JOYCE site was funded by the Transregional Collaborative Research Center TR32 (Simmer et al., 2015) funded by DFG. We thank the three anonymous reviewers for their detailed and thorough comments which greatly helped to improve the manuscript.

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
