# Peer review of "How to estimate total differential attenuation due to hydrometeors with ground-based multi-frequency radars?"

_Atmospheric Measurement Techniques, 2020_

## Referee Comment (RC1) · Anonymous Referee #1 · 16 Jun 2020

This paper discusses a new approach to estimate the total differential attenuation in cloud/precipitation radar profiles, a key constraint for attenuation correction, which is needed with multi-frequency radars using higher frequencies. The authors use the regions of small hydrometeors at the top of the cloud-precipitation column, where different frequencies should have the same radar reflectivity, to derive the total differential attenuation. They propose a robust way to estimate where the small hydrometeors are located based on the vertical gradient of the dual-frequency ratio.

This paper is well written and clear, with good visualizations to illustrate the method used. The proposed technique shows promise to be a significant contribution to dealing

with the issue of (differential) attenuation. I only have a few comments and suggestions for improvement and thus recommend to accept the paper with minor revisions.

General comments:

1. The paper could use some discussion of the applicability of the limitations of the method:

1a: Most importantly, almost all the discussion seems to implicitly assume that the radar is zenith-pointing, but this is not actually mentioned (as far as I can see) in the text until line 164. Do you expect the method to be applicable to non-vertically pointing radars?

1b: Also, are there conditions where the algorithm will/might fail? For example, if there is very heavy attenuation in the lower part of the column, I imagine that this might prevent the radar from detecting the small-particle region altogether. Multiple scattering might also be an issue in such cases. I'm not demanding that the authors solve all these problems in this paper, but they should at least be discussed because they are important issues to deal with if this algorithm is ever to be used in an automated or semi-automated fashion to process large datasets.

2. The authors promote the method as an improvement compared to the earlier technique of estimating the baseline differential attenuation from low-reflectivity regions. It would make this paper more convincing if they actually compared the (quite impressive) results obtained with their method to those obtained with the older method. For example, Fig. 5c would be a good place to put such a comparison.

Specific comments:

Line 91: "While attenuation mainly limits the maximum range of possible radar observations": Doesn't it also introduce errors to the retrievals because you have more uncertainty in the reflectivity?

Equation 1: "cw" here means cloud water? Please specify.

Figure 1 caption: I realize that 1 $\mu$m is used here as a "small-particle limit" but it seems a little odd given that 1 $\mu$m droplets aren't even stable (also, it's not mentioned if 1 $\mu$m is the radius or the diameter...)

Line 151: How is EWC defined?

Lines 209-210: Related to my general comment #2 above, what do you do if a Rayleigh plateau satisfying the conditions is not found?

Discussion of Fig. 5: You should discuss a little bit the apparent negative PIAs in Fig. 5c (e.g. between 05:00 and 05:20). This is surely not physical. What causes these artifacts and how do you handle them?

Figure 6: The circles seem to overlap each other quite a bit here, using different size or shape markers might be better.

Figure 7c: Here we see not only negative PIA but also negative LWP in the first minutes of the time series. Why?

---

## Referee Comment (RC2) · Anonymous Referee #2 · 16 Jun 2020

General comments:

The authors present a handy method to identify regions where hydrometeors can be assumed as Rayleigh scatterers at Ka and W bands. This Rayleigh region is useful, because it allows the retrieval of the path-integrated differential attenuation between Ka and W bands. Then, the authors elaborate how the derived differential attenuation can be used to estimate the liquid water path. The method presented is independent of a threshold reflectivity as was used in previous studies and has potential applications in multi-frequency radar observations.

The authors well illustrate the background of this study and cite relevant literature. This

manuscript has some interesting aspects which deserve publication. However, more clarifications are still needed. Please see my comments below.

Major comments:

1) The title confuses me. I believe the community has already been aware of the answer to this question, namely, matching reflectivities at the cloud top facilitates the retrieval of total differential attenuation. After reading this manuscript, I feel that the most innovative part is the presented method for identifying the Rayleigh region in clouds and its applications. I suggest the authors modify the title based on their main contribution.

2) Although the gaseous attenuation has been noted in Section 2, it is necessary to elaborate its impact in a subsection. I also have two questions. 1) the gaseous attenuation has been corrected in the BAECC case, have you done the calibration for the TRIPEx case? 2) How would the performance of this method be affected without correcting the gaseous attenuation? I believe it will at least modify the DFR profile.

3) I think the method in general works well. The edges of the non-Rayleigh areas in Figure 5b 7b are more or less smooth, which is reasonable and expected. However, they are rather noisy in Figure 10 b (many spikes). Those spikes can be troublesome in applications and indicate the technical limitations of this method. But such information seems missing in the manuscript. In particular, there should be one section describing the conditions that this method is applicable. At least, the scenario of rain can be problematic.

Minor comments:

1) I suggest the use of DWR instead of DFR, since the DWR is more widely used in the community.

2) Equation 2. Could you please specify the meaning of equivalent water content and its unit?

3) L155. It is better to specify why not water droplets.

4) L178 and L190. The cloud top can be covered by a layer of liquid. The dielectric constants of liquid water are different at Ka and W bands, then the observed DFR is different even for Rayleigh-scattering liquid drops. Given the method is mainly applied to cloud top where a layer of liquid is commonly observed, how did you exclude the existence of this liquid layer? Maybe it is not easy to recognize this liquid layer without using lidar measurements, then how the DFR would be affected without the information about the hydrometeor phase? I expect it to be relatively small, have you quantified it?

5) Figure 2. It takes a lot effort for me to match the block in the flow chart to the explanation in the text. It will be more readable if you could number each block in the flow chart and order the explanation by serial numbers.

6) Figure 3. It helps the interpretation if you could also present the reflectivity profiles at Ka and W band

7) L207. Why the Savitzky-Golay filter is used? It works much better than other filters?

8) Figure 4b. Why there is a spike of DFR up to 9 km at around 4:20? Strong liquid attenuation? This spike seems resulting in the misclassification of non-Rayleigh region in Figure 5b. Could you please elaborate the reason and hint the readers the limitation of this method?

9) L256. Why the opposite is expected? Because of the melting?

10) Figure 5. The layout of (c) should be improved to match with (a) and (b).

11) L275. Although the agreement looks good, I am curious how much attenuation can be attributed to ice attenuation. It seems to me that the Ka-band reflectivity and DFR(Ka,W) are not that small.

12) Figure 6. It is hard for me to recognize the periods. Given the interrupts by rain, I suggest the authors mark the same short period by one color range.

13) L280. Radome attenuation should also affect the $\Delta$PIA.

14) Figure 7 & 8. This case well demonstrates the dependence of liquid attenuation on the temperature. There is no liquid cloud below 4 km, therefore the liquid layer should be detected by the ceilometer. It would strengthen the conclusion if the cloud base detected by the ceilometer is marked in Figure 7(b).

15) L358. Why this temperature region is expected? Have you checked the lidar data? Marking the liquid layer in the plot will be more convincing.

16) L375. Are you assuming the temperature of -10 deg?

17) L394. What is the maximum measurable LWP for MWR? At around 2:30, the agreement seems rather good although the LWP is large.

18) L408. 'negligible particle growth'. odd statement. What matters is the particle size instead of its growth.

Typos: 1) L80: 'non-perfect' 2) L116: 'higher frequencies' 3) L140: 'W-band' is missing 4) L170: 'generally' 5) L234: 'by Dias Neto et al.' 6) L295: not 'curve' in Fig.6 7) L329: '(Kalesse et al., 2016)'

———————————————

---

## Referee Comment (RC3) · Anonymous Referee #3 · 17 Jun 2020

This is a clearly written and carefully presented manuscript describing a novel and useful method for processing zenith-pointing ground based multiple-frequency radar observations, a configuration used across many field sites, which produce important and ongoing data series. The automated method for identifying the "Rayleigh plateau" in multiple-frequency radar reflectivity profiles of clouds reduces major uncerainties inherent in simpler threshold-based methods, while demonstrably increasing the number of gates identified as containing Rayleigh scattering ice, a result I would like to see better quantified in this paper. The estimated path-integrated attenuation based on this method compares very favourably to liquid water path estimates from microwave radiometers, and the authors point out the potential of this methodology to form the basis
of a profiling liquid water content retrieval in synergy with microwave radiometers and additional radar frequencies.

Subject to some minor revisions, I recommend the paper for publication.

Major comments:

In the time-height plots of Z and DFR for the two cases (Figs 5 & 10), the shading and black contour show the difference between a Z < -10 dBZ threshold and their method. It seems that this under-plays what should be a major result of the method described in this paper. Could you quantify the fractional or absolute difference between the two methods for identifying the Rayleigh plateau? Building on the very clear discussion in the introduction, it would be good to quantify not just the additional gates gained, but also the gates that would have been treated as Rayleigh scattering by a threshold method, and which the new method can identify as containing a small number of larger ice particles.

Minor comments:

- P2, L22-3 "...but also for differences in models..." is a subclause, and need some punctuation.

- Fig. 1: the legend uses GHz definitions for radars frequencies, rather than the Ka- / W- / G-band nomenclatures used throughout the paper. It's worth being consistent.

- Fig. 3 & L215–6. Best be clear that the "very low reflectivities" here are at the Ka-band. These first examples of the method might be illustrated more clearly by including an additional panel showing the Ka and W-band radar reflectivities for these profiles, then the DFR and the gradient of DFR, rather than referring the reader to Fig. 9.

- Figs. 5 & 10. It seems a small thing, but it greatly helps interpretation of these time series figures if the x-axes of all panels are aligned.

- L278–280: "As seen in Fig. 6, no LWP is derived during rainy periods (before 01:00,

between 07:00and 08:00, 9:00 and 13:00 and after 16:00 UTC)..." Should this refer instead to Fig. 7c?

- L390–97: It's worth pointing out both possibilities for the mismatch in attenuation; however, does the extensive multiple-frequency Doppler radar literature on this case suggest one is more likely than another?

- Fig. 7, P13, L294-7; can you please clarify in the text and the caption of Figure 7 if the same temperature is assumed in Fig. 7 as in Fig. 5?

---

## Author Response (AR1)

**Responses to reviews**

How to estimate total differential attenuation due to hydrometeors with ground-based multi-frequency radars?

F. Tridon, A. Battaglia, S. Kneifel

June 29th, 2020

We thank all three reviewers for their efforts and time for reviewing as well as constructive comments which greatly helped to improve the manuscript. All our point-to-point answers are highlighted in red below according to the following sequence: (i) comments from referees/public, (ii) author's response, and (iii) author's changes in manuscript.

**Comments from reviewer 1**

This paper discusses a new approach to estimate the total differential attenuation in cloud/precipitation radar profiles, a key constraint for attenuation correction, which is needed with multi-frequency radars using higher frequencies. The authors use the regions of small hydrometeors at the top of the cloud-precipitation column, where different frequencies should have the same radar reflectivity, to derive the total differential attenuation. They propose a robust way to estimate where the small hydrometeors are located based on the vertical gradient of the dual-frequency ratio.

This paper is well written and clear, with good visualizations to illustrate the method used. The proposed technique shows promise to be a significant contribution to dealing with the issue of (differential) attenuation. I only have a few comments and suggestions for improvement and thus recommend to accept the paper with minor revisions.

**General comments:**

1. The paper could use some discussion of the applicability of the limitations of the method:

1a: (i) Most importantly, almost all the discussion seems to implicitly assume that the radar is zenith-pointing, but this is not actually mentioned (as far as I can see) in the

text until line 164. Do you expect the method to be applicable to non-vertically pointing radars?

(ii) Indeed, this important information was missing. Thanks for the comment. Yes of course, when scanning radars are available, the method can be applied to slant profiles. This would even extend the applicability of the method to cases where attenuation is weaker (e.g. thin liquid cloud layer) because the signal would be enhanced.

(iii) We now mention this information at the end of section 1 and in the conclusion.

1b: (i) Also, are there conditions where the algorithm will/might fail? For example, if there is very heavy attenuation in the lower part of the column, I imagine that this might prevent the radar from detecting the small-particle region altogether. Multiple scattering might also be an issue in such cases. I'm not demanding that the authors solve all these problems in this paper, but they should at least be discussed because they are important issues to deal with if this algorithm is ever to be used in an automated or semi-automated fashion to process large datasets.

(ii) There are clearly some regions where no Rayleigh plateau can be found. It can indeed happen in case of heavy attenuation due to rain for example, which would make the radar not sensitive enough to detect the small-particle region. More generally, it might be difficult to find a clear Rayleigh plateau when cloud tops are irregular and jagged. For example, this is actually the case for the second case study (Section 5) between 15h and 16h30 UTC during which no DeltaPIA can be retrieved (no data points during this period in Fig. 11).
On the contrary, as described in Battaglia et al. (2010), the critical condition to lead to multiple scattering is when the mean free-radiation path (defined as the inverse of the extinction coefficient) is comparable or lower than the beamwidth of the radar. Ground-based high-frequency cloud radars usually have such a narrow beamwidth that multiple scattering is seldom a problem. It can be easily verified by looking at the linear depolarization ratio which would increase substantially in case of multiple scattering (which is not found for the cases analyzed in our study).
(iii) We added this information in the new section 3.3 and completed the flowchart in Fig. 2.
Reference:
        Battaglia, A., S. Tanelli, S. Kobayashi, D. Zrnic, R.J. Hogan, C. Simmer, Multiple-scattering in radar systems: a review, *J. Quant. Spec. Rad. Transf.*, 2010, 111 (6), 917-947.

2. (i) The authors promote the method as an improvement compared to the earlier technique of estimating the baseline differential attenuation from low-reflectivity regions. It would make this paper more convincing if they actually compared the (quite impressive) results obtained with their method to those obtained with the older method. For example, Fig. 5c would be a good place to put such a comparison.

(ii), (iii) We have now added a full comparison of the results obtained with the Rayleigh plateau and Z-threshold methods, including some statistics, with updated

figures 5c, 6, 7c, 8, 10c, 11. The quantitative improvements compared to the older method are not striking in terms of LWP retrieval. However, as we described in the initial version of the paper (conclusion), the main advantages of the new method are 1) that it can be applied independently of the radar frequency pair (without the need of fine tuning a Z threshold), 2) it exploits a much a larger region (which in general should lead to a better accuracy) and 3) provides quality controlled estimates (no DeltaPIA can be retrieved if no Rayleigh plateau is found) whereas the threshold method has intrinsically no DFR quality check. We emphasized these points now in the conclusion.

**Specific comments:**

1) Line 91: (i) "While attenuation mainly limits the maximum range of possible radar observations": Doesn't it also introduce errors to the retrievals because you have more uncertainty in the reflectivity?

(ii) Yes indeed, this is why we used the word "mainly". Attenuation leads to a decrease of signal to noise ratio (SNR), and hence, a larger uncertainty in reflectivity and in the DeltaPIA estimate. We think that mentioning this technical issue at this place would divert the main message which is that the attenuation signal can be used as a source of information. (iii) Instead, in the description of the algorithm, we added an item about the lower SNR limit that we use for keeping only reliable reflectivity data and we added this filtering explicitly in the flowchart of Fig. 2.

2) Equation 1: (i) "cw" here means cloud water? Please specify.
(ii) Yes. (iii) Done.

3) Figure 1 caption: (i) I realize that 1 m is used here as a "small-particle limit" but it seems a little odd given that 1 m droplets aren't even stable (also, it's not mentioned if 1 m is the radius or the diameter...)

(ii) (iii) Indeed, we changed the legend and the figure using droplets of 10 µm radius (even at G-band the change is almost invisible). Anyhow the result is independent to the selection of radius as far as it is much smaller than the radar wavelength.

4) Line 151: (i) How is EWC defined?
(ii) EWC is simply the water content in g/m^3. (iii) We clarified this in the manuscript.

5) Lines 209-210: (i) Related to my general comment #2 above, what do you do if a Rayleigh plateau satisfying the conditions is not found?

(ii) When no Rayleigh plateau can be found while a cloud is present (i.e. significant reflectivity is found in the profile), the DeltaPIA is simply set to "missing value". (iii) This has been added in the text.

6) Discussion of Fig. 5: (i) You should discuss a little bit the apparent negative PIAs in Fig. 5c (e.g. between 05:00 and 05:20). This is surely not physical. What causes these artifacts and how do you handle them?

(ii), (iii) These negative PIA where mainly coming from a slight bias (0.24 dB) in the relative calibration of the two radars. Indeed, in the initial version of the manuscript we assumed that attenuation produced by the ice cloud for this case was negligible (as it is commonly done) and used the DeltaPIAs obtained when only LWP is negligible (left panel in the figure below suggests a "relative calibration constant" of 5.7 dB) to adjust the reflectivity of the two radars (in this way, the resulting DeltaPIA is equal to zero in average). In reality, it seems that the thick ice cloud produces some attenuation at W-band: when also removing data associated with large IWP (computed from $Z_{Ka}$ following the procedure of section 5.2), the "relative calibration constant" reduces to 5.46 dB (right panel in the figure below). With this new calibration, the retrieved DeltaPIA is raised overall by 0.24 dB, which removes most of the negative PIAs in Fig. 5c. Since the ice attenuation is much smaller than liquid attenuation for this case, we decided to keep the direct comparison between DeltaPIA and LWP. As a result, we find a slight positive bias which can be explained by ice attenuation, as we now mention in the manuscript.

[Figure]

Comparison of reflectivities measured by both radars within the Rayleigh plateau areas where (left) LWP < 40g/m$^2$ (data used for calibration in submitted manuscript) and where (right) LWP < 40g/m$^2$ and IWP < 500g/m$^2$ (data used for calibration in the new version).

Of course, the remaining negative PIAs can be due to the random error in reflectivity measurements at low SNR. This is particularly likely when few measurements at cloud top are used and probably the case for the Z-threshold method between 5:00 and 5:30 UTC in Fig. 5, as we now mention in the manuscript, in the description of Fig. 5c.

In order to avoid as much as possible negative DeltaPIA retrievals, we apply a 20 s moving average for both methods.

7) Figure 6: (i) The circles seem to overlap each other quite a bit here, using different size or shape markers might be better.

(ii) (iii) The figure has been updated with smaller makers.

8) Figure 7c: (i) Here we see not only negative PIA but also negative LWP in the first minutes of the time series. Why?

(ii) For the TRIPEX case study, we use LWP obtained from a statistical retrieval, which is based on a regression over a large amount of data (see Löhnert and Crewell, 2003, cited in the manuscript). It is therefore not surprising that it sometimes gives unphysical negative values. (iii) We now mention this in the manuscript.

**Comments from reviewer 2**

 The authors present a handy method to identify regions where hydrometeors can be assumed as Rayleigh scatterers at Ka and W bands. This Rayleigh region is useful, because it allows the retrieval of the path-integrated differential attenuation between Ka and W bands. Then, the authors elaborate how the derived differential attenuation can be used to estimate the liquid water path. The method presented is independent of a threshold reflectivity as was used in previous studies and has potential applications in multi-frequency radar observations.

The authors well illustrate the background of this study and cite relevant literature. This manuscript has some interesting aspects which deserve publication. However, more clarifications are still needed. Please see my comments below.

**Major comments:**

1) (i) The title confuses me. I believe the community has already been aware of the answer to this question, namely, matching reflectivities at the cloud top facilitates the retrieval of total differential attenuation. After reading this manuscript, I feel that the most innovative part is the presented method for identifying the Rayleigh region in clouds and its applications. I suggest the authors modify the title based on their main contribution.

(ii) (iii) We changed the title to "Estimating total attenuation using Rayleigh targets at cloud top: applications in multi-layer and mixed-phase clouds observed by ground-based multi-frequency radars".

2) (i) Although the gaseous attenuation has been noted in Section 2, it is necessary to elaborate its impact in a subsection. I also have two questions. 1) the gaseous attenuation has been corrected in the BAECC case, have you done the calibration for the TRIPEx case? 2) How would the performance of this method be affected without correcting the gaseous attenuation? I believe it will at least modify the DFR profile.

(ii) 1) In the TRIPEx level 2 data that we use, the gaseous attenuation correction is already applied. 2) Gaseous attenuation must indeed be applied. It is significantly larger at W-band than at Ka-band: average value of total gas attenuation at cloud tops are 0.5dB and 1.4dB for the TRIPEX case and 0.45dB and 1.15dB for the BAECC case, at Ka and W-band, respectively. In principle, even if we would not correct for it beforehand, the relative calibration at cloud top would also compensate for the total gas attenuation. The figure below (to be compared with Fig. 3 of the manuscript) shows the resulting profiles without gas attenuation correction. There is

indeed a very small difference which is explained because relative calibration is performed from data at the beginning of the case (where LWP and IWP are small) where gas attenuation is slightly smaller (due to variations in thermodynamic properties over time during the case study).

[Figure]

Same as Fig. 3 of the manuscript but where the reflectivity has not been corrected from gas attenuation. The profiles are only slightly different (Zw slightly lower and DWR slightly larger). The DFR gradient regions are also slightly changed but this has little effect on the resulting DeltaPIA since the median of the Rayleigh plateau is retained.

Another effect is indeed the modification of the slope of the DFR profile. The majority of total atmospheric water vapor - which is the main contributor to gas attenuation - is located in the boundary layer and maybe a few km above it. In fact, in our cases, we find 50% of the gas attenuation to occur in the lowest 2km. So we are very convinced that when the method is applied for larger heights, the effect of a missing or wrong slope in DWR due to gas attenuation would be very small if not even negligible. Only if the technique would be applied to low layer clouds, such as Arctic mixed-phase clouds, we would imagine to see any effect. However, for liquid topped mixed-phase clouds, the primary limitation of the method would be the presence of liquid layer at cloud top (as it is now discussed in section 3.3).

(iii) There is no question about the fact that gas attenuation must be corrected in the first place. A question which can be raised is the uncertainty of the model for computing the amount of gas attenuation but this error is small. Therefore, instead of a full new section, we added a note of caution in the manuscript about case studies where it could play a role (e.g. boundary layer clouds) even if the resulting change of slope would be minor compared to other limitations of the method.
Furthermore, we added the information in the description of the technique (and in the flowchart of Fig. 2) that gaseous attenuation must be corrected for and that it is already corrected in the TRIPEx dataset that we use.

3) (i) I think the method in general works well. The edges of the non-Rayleigh areas in Figure 5b 7b are more or less smooth, which is reasonable and expected.

However, they are rather noisy in Figure 10 b (many spikes). Those spikes can be troublesome in applications and indicate the technical limitations of this method. But such information seems missing in the manuscript. In particular, there should be one section describing the conditions that this method is applicable. At least, the scenario of rain can be problematic.

(ii) The spikes in Rayleigh plateau detection are not surprising: they come from the absolute threshold on the maximum DFR gradient. Such threshold can be exceeded or not in two consecutive and similar profiles and the length of the resulting Rayleigh plateau can then be very different. In our opinion, obtaining a rather continuous DeltaPIA with discontinuous Rayleigh plateau detection (e.g. between 22:00 and 22:30 UTC in Fig. 10) is actually an indication that the method works well. An easy way to avoid such discontinuities would be to smooth the flag of Rayleigh plateau detection but we prefer to keep showing the unsmoothed result of the Rayleigh plateau detection so that one can openly judge the results. (iii) On the other hand, there are indeed some profiles where no Rayleigh plateau can be detected (c.f. reply to Reviewer 1 comment 1b) and this is now described in paragraph 3.3.

**Minor comments:**

1) (i) I suggest the use of DWR instead of DFR, since the DWR is more widely used in the community.

(ii) (iii) We thank the reviewer for the suggestion but we think that both ways are widely used in the community (see for example papers related to GPM) so we prefer to stick to DFR.

2) (i) Equation 2. Could you please specify the meaning of equivalent water content and its unit?

(ii) EWC is simply the water content in g/m^3. (iii) We clarified this in the manuscript.

3) (i) L155. It is better to specify why not water droplets.

(ii) We made this choice because we expect that in the majority of clouds (one exception is mixed-phase clouds as discussed in the next comment), the reflectivity of cloud tops is dominated by ice particles. (iii) This is now clearly described in the new version of the manuscript.

4) (i) L178 and L190. The cloud top can be covered by a layer of liquid. The dielectric constants of liquid water are different at Ka and W bands, then the observed DFR is different even for Rayleigh-scattering liquid drops. Given the method is mainly applied to cloud top where a layer of liquid is commonly observed, how did you exclude the existence of this liquid layer? Maybe it is not easy to recognize this liquid

layer without using lidar measurements, then how the DFR would be affected without the information about the hydrometeor phase? I expect it to be relatively small, have you quantified it?

(ii) In case of liquid-only cloud top, the DeltaPIA would be overestimated by about 1 dB for the Ka-W band pair at 0°C due to the dielectric constant variation as a function of frequency. However, the presence of a few ice crystals would also largely influence the reflectivity since they would dominate the signal and thus the net effect would be lower. In a single layer mixed-phase cloud, it is indeed a limitation of the algorithm which could only be solved by exploiting Doppler spectra to separate liquid and ice contributions. Note, however, that in such a situation, it would not be possible to find Rayleigh plateau and the method would not provide any erroneous DeltaPIA. (iii) Discussion of this limitation has been added in the new section 3.3.

5) (i) Figure 2. It takes a lot effort for me to match the block in the flow chart to the explanation in the text. It will be more readable if you could number each block in the flow chart and order the explanation by serial numbers.

(ii) (iii) Instead of adding numbers to the blocks (which sounds a bit unconventional for a flowchart), we decided to mark the different filtering steps in the text.

6) (i) Figure 3. It helps the interpretation if you could also present the reflectivity profiles at Ka and W band

(ii) (iii) Done.

7) (i) L207. Why the Savitzky-Golay filter is used? It works much better than other filters?

(ii) (iii) There is indeed no specific reason for using the Savitzky-Golay filter so we simplified the text and the flowchart.

8) (i) Figure 4b. Why there is a spike of DFR up to 9 km at around 4:20? Strong liquid attenuation? This spike seems resulting in the misclassification of non-Rayleigh region in Figure 5b. Could you please elaborate the reason and hint the readers the limitation of this method?

(ii) We are not sure if we know which DFR spike the reviewer is referring to in figure 4b: there is indeed the peak (actually multiple peaks) of attenuation produced by the cumulus clouds around 04:45 (as better seen in Fig. 5b). On the contrary, there is indeed a small period around 04:20 where no Rayleigh plateau can be found within the whole profile. The reason for this is an unfortunate combination of low $SNR_W$ and high variance of DFR near cloud top leading to the filtering of all data down to 1 km

from cloud top (see figure below). The condition that a Rayleigh plateau must be located at less than 500m from cloud top is then not satisfied.

[Figure]

Same as Fig. 5b of the manuscript but where the shown DFR has been pre-filtered according to the criteria described in section 3.1. As a result, the gray areas correspond to low $SNR_W$, high variance of DFR, high $Z_{Ka}$ or high variance of $Z_{Ka}$ while the gray shading on top of colored DFR corresponds to areas which have not been identified as a Rayleigh plateau. Around 4:20 UTC, the highest usable DFR is at around 9 km while the cloud top is at around 10 km.

(iii) This feature is the result of tradeoffs the algorithm has to deal with: we don't want to relax this condition as this could lead to an underestimation of the DeltaPIA in case of attenuation produced near cloud top. Of course, there will be some profiles where the algorithm cannot work but we don't want to put the focus on such a specific feature in section 4.2 as it would divert the reader from the main message. Instead, there is now a discussion on general limitations of the algorithm in a new paragraph according to the reviewer's comment #3.

9) (i) L256. Why the opposite is expected? Because of the melting?

(ii) Z should increase downward if it were an ice cloud because ice crystals are growing while falling. But indeed, another reason is that the reflectivity of droplets would dramatically decrease while freezing. (iii) We added this new argument in the manuscript.

10) (i) Figure 5. The layout of (c) should be improved to match with (a) and (b).

(ii) (iii) Done.

11) (i) L275. Although the agreement looks good, I am curious how much attenuation can be attributed to ice attenuation. It seems to me that the Ka-band reflectivity and DFR(Ka,W) are not that small.

(ii) By solving the issue of negative DeltaPIA pointed out by Reviewer 1 minor comment 6, we realized that the ice clouds produce indeed about 0.2 dB attenuation on average. (iii) Even if this is a rather small value, we updated the relative

calibration by using only data where IWP is expected to be small (see our reply to the reviewer 1). As a result, the retrieved DeltaPIA is raised by 0.2 dB for the whole case study. DeltaPIA now appears to be a bit too large compared to the measured LWP, which can be explained by ice attenuation, as we now mention in the manuscript.

12) (i) Figure 6. It is hard for me to recognize the periods. Given the interrupts by rain, I suggest the authors mark the same short period by one color range.

(ii) (iii) We have updated the figure which now uses 2 colorbars. It allows to clearly see the variations between 04:00 and 05:30 UTC while presenting the data for the whole case study.

13) (i) L280. Radome attenuation should also affect the PIA.

(ii) (iii) We now mention that a blower has to be used in order to avoid the issue of wet radome attenuation.

14) (i) Figure 7 & 8. This case well demonstrates the dependence of liquid attenuation on the temperature. There is no liquid cloud below 4 km, therefore the liquid layer should be detected by the ceilometer. It would strengthen the conclusion if the cloud base detected by the ceilometer is marked in Figure 7(b).

(ii) (iii) Indeed we have now added the liquid cloud base as detected by the ceilometer in Z_ka panels of Fig. 5 and 7. The corresponding lidar backscatter and cloud base detection is also shown in the plots below. This confirms the presence of a supercooled liquid water cloud at 5 km between 15:30 and 15:45 UTC.

[Figure]

[Figure]

Lidar backscatter corresponding to the plots of figures 5 and 7. The black dots denote the detection of a liquid cloud base.

15) (i) L358. Why this temperature region is expected? Have you checked the lidar data? Marking the liquid layer in the plot will be more convincing.

(ii) We estimated roughly the temperature range from radiosoundings measurements at 17:00 and 23:00 UTC which suggest saturation with respect to water in this range of temperature. Unfortunately, the lidar signal is already completely extinguished from 1km for most of the case study (see figure below) so it does not help for guessing the averaged height of the liquid cloud. (iii) We added the information on radiosoundings and the cloud base detected by the lidar in Fig. 9a and 10a.

[Figure]

Measured lidar backscatter measured within the same time and height limits as Fig. 9 of the manuscript.

16) (i) L375. Are you assuming the temperature of -10 deg?

(ii) (iii) Yes, we added the information in the manuscript.

17) (i) L394. What is the maximum measurable LWP for MWR? At around 2:30, the agreement seems rather good although the LWP is large.

(ii) There is no real limitation as long as there is no drizzle drop which leads to larger differential attenuation per unit mass. (iii) However, we think that the best explanation for the mismatch found between retrieved snow attenuation and IWP is the possibility that aggregates are significantly rimed. We added this information in the manuscript.

18) (i) L408. 'negligible particle growth'. odd statement. What matters is the particle size instead of its growth.

(ii) (iii) This has been rephrased.

Typos: (i) 1) L80: 'non-perfect' 2) L116: 'higher frequencies' 3) L140: 'W-band' is missing 4) L170: 'generally' 5) L234: 'by Dias Neto et al.' 6) L295: not 'curve' in Fig.6 7) L329: '(Kalesse et al., 2016)'

(ii) (iii) Done. Thanks for the thorough proofread.

**Comments from reviewer 3**

This is a clearly written and carefully presented manuscript describing a novel and useful method for processing zenith-pointing ground based multiple-frequency radar observations, a configuration used across many field sites, which produce important and ongoing data series. The automated method for identifying the "Rayleigh plateau" in multiple-frequency radar reflectivity profiles of clouds reduces major uncertainties inherent in simpler threshold-based methods, while demonstrably increasing the number of gates identified as containing Rayleigh scattering ice, a result I would like to see better quantified in this paper. The estimated path-integrated attenuation based on this method compares very favourably to liquid water path estimates from microwave radiometers, and the authors point out the potential of this methodology to form the basis of a profiling liquid water content retrieval in synergy with microwave radiometers and additional radar frequencies.

**Major comment:**

(i) In the time-height plots of Z and DFR for the two cases (Figs 5 & 10), the shading and black contour show the difference between a Z < -10 dBZ threshold and their method. It seems that this under-plays what should be a major result of the method described in this paper. Could you quantify the fractional or absolute difference between the two methods for identifying the Rayleigh plateau? Building on the very clear discussion in the introduction, it would be good to quantify not just the additional gates gained, but also the gates that would have been treated as Rayleigh scattering by a threshold method, and which the new method can identify as containing a small number of larger ice particles.

(ii) Because of the characteristics of the two case studies (e.g. upper cirrus cloud in the BAECC case), the threshold method very rarely mis-classifies non-Rayleigh regions. Nevertheless, the Rayleigh plateau method shows better performances statistically. (iii) We are now discussing the few instances of mis-classification by the threshold method in Fig.10. And in particular, we have now added a full comparison of the results obtained with the Rayleigh plateau and Z-threshold methods, including some statistics, with updated figures 5c, 6, 7c, 8, 10c, 11. The quantitative improvement compared to the older method are not very large in terms of LWP retrieval. However, as we described in the initial version of the paper (conclusion), the main advantage of the new method is that it can be applied 1) independently of the radar frequency pair (without the need of fine tuning a Z threshold), 2) it exploits a much a larger region (which in general should lead to a better accuracy) and 3) it provides quality controlled estimates (no DeltaPIA can be retrieved if no Rayleigh plateau is found). We emphasized these items in the conclusion.

**Minor comments:**

1) (i) P2, L22-3 "...but also for differences in models: : :" is a subclause, and need some punctuation.

(ii) (iii) Done.

2) (i) Fig. 1: the legend uses GHz definitions for radars frequencies, rather than the Ka- / W- / G-band nomenclatures used throughout the paper. It's worth being consistent.

(ii) (iii) Done (the legend has been updated with radar bands)

3) (i) Fig. 3 & L215–6. Best be clear that the "very low reflectivities" here are at the Kaband. These first examples of the method might be illustrated more clearly by including an additional panel showing the Ka and W-band radar reflectivities for these profiles, then the DFR and the gradient of DFR, rather than referring the reader to Fig. 9.

(ii) (iii) Done.

4) (i) Figs. 5 & 10. It seems a small thing, but it greatly helps interpretation of these time series figures if the x-axes of all panels are aligned.

(ii) (iii) Done.

5) (i) L278–280: "As seen in Fig. 6, no LWP is derived during rainy periods (before 01:00, between 07:00and 08:00, 9:00 and 13:00 and after 16:00 UTC)..." Should this refer instead to Fig. 7c?

(ii) No we were referring to the lack of points in the scatterplot during these time periods. (iii) This has been clarified.

6) (i) L390–97: It's worth pointing out both possibilities for the mismatch in attenuation; however, does the extensive multiple-frequency Doppler radar literature on this case suggest one is more likely than another?

(ii) The literature suggests that snow bulk density is particularly large during those periods. (iii) This has been added in the manuscript.

7) (i) Fig. 7, P13, L294-7; can you please clarify in the text and the caption of Figure 7 if the same temperature is assumed in Fig. 7 as in Fig. 5?

(ii) Yes indeed, the Y-axis scale follows the same convention as in Fig. 5, for consistency. (iii) We have clarified this in both the text and caption.

[revised manuscript text omitted]